# Cardiologist-level interpretable knowledge-fused deep neural network for automatic arrhythmia diagnosis
Yanrui Jin [1,2,4], Zhiyuan Li[1,2,4], Mengxiao Wang[1,2], Jinlei Liu[1,2], Yuanyuan Tian[1,2], Yunqing Liu[1,2], Xiaoyang Wei[1,2], Liqun Zhao[3,5] ✉ & Chengliang Liu [1,2,5] ✉

## Abstract

**Background** Long-term monitoring of Electrocardiogram (ECG) recordings is crucial to diagnose arrhythmias. Clinicians can find it challenging to diagnose arrhythmias, and this is a particular issue in more remote and underdeveloped areas. The development of digital ECG and AI methods could assist clinicians who need to diagnose arrhythmias outside of the hospital setting.
**Methods** We constructed a large-scale Chinese ECG benchmark dataset using data from 272,753 patients collected from January 2017 to December 2021. The dataset contains ECG recordings from all common arrhythmias present in the Chinese population. Several experienced cardiologists from Shanghai First People's Hospital labeled the dataset. We then developed a deep learning-based multi-label interpretable diagnostic model from the ECG recordings. We utilized Accuracy, F1 score and AUC-ROC to compare the performance of our model with that of the cardiologists, as well as with six comparison models, using testing and hidden data sets.
**Results** The results show that our approach achieves an F1 score of 83.51%, an average AUC ROC score of 0.977, and 93.74% mean accuracy for 6 common arrhythmias. Results from the hidden dataset demonstrate the performance of our approach exceeds that of cardiologists. Our approach also highlights the diagnostic process.
**Conclusions** Our diagnosis system has superior diagnostic performance over that of clinicians. It also has the potential to help clinicians rapidly identify abnormal regions on ECG recordings, thus improving efficiency and accuracy of clinical ECG diagnosis in China. This approach could therefore potentially improve the productivity of out-of-hospital ECG diagnosis and provides a promising prospect for telemedicine.

## Plain language summary

Arrhythmia, also known as an irregular heartbeat, is a common cardiovascular disease. Sometimes the presence of an arrhythmia can increase the risk of more serious heart conditions. Long-term monitoring of the heartbeat enables arrhythmia to be more easily diagnosed. To accurately detect arrhythmia, we developed a computational model that was able to detect six common types of arrhythmias from readings of the heart rate obtained using a device connected to a mobile phone. We showed that our model could diagnose these arrhythmias in over 270,000 people living in China. Our diagnostic system could enable arrhythmias to be diagnosed more easily outside of hospitals and therefore improve access to healthcare, particularly for those in remote settings.

More than 544, 000 cardiovascular-related deaths occur yearly in China[1], but less than 1% of the cases can receive prompt rescue, which severely poses challenges in timely and effective treatment. Arrhythmia is the leading cause of cardiovascular death, and timely arrhythmia detection is of great significance to the prevention of sudden cardiac death. Clinically, Electrocardiogram (ECG) is the most effective and straightforward clinical technique to detect arrhythmias. The process of analyzing ECG is meticulous, time-consuming, and empirical in nature, necessitating the involvement of numerous professional clinicians to meet daily requirements. However, remote and underdeveloped areas lack sufficient professional clinicians, thereby aggravating the scarcity of medical resources[2]. Recently, the rapid development of telemedicine presents an effective solution to ease healthcare resource inadequacies in remote and underdeveloped regions. Consequently, considering the huge diagnostic demands

[1]State Key Laboratory of Mechanical System and Vibration, School of Mechanical Engineering, Shanghai Jiao Tong University, Shanghai, China. [2]MoE Key Lab of Artificial Intelligence, AI Institute, Shanghai Jiao Tong University, Shanghai, China. [3]Department of cardiology, Shanghai First People's Hospital Affiliated to Shanghai Jiao Tong University, Shanghai, China. [4]These authors contributed equally: Yanrui Jin, Zhiyuan Li. [5]These authors jointly supervised this work: Liqun Zhao, Chengliang Liu. ✉e-mail: zhaolq8570@hotmail.com; chlliu@sjtu.edu.cn

of ECG and the developmental demands of telemedicine, the creation of a reliable and accurate automatic arrhythmia diagnosis system is a pressing priority in advancing smart healthcare technology.

Traditional diagnosis approaches[3–8] retain interpretability with a medical theoretical foundation, but they are not very robust and are not appropriate for clinical application due to two limitations. Firstly, the model performance depends on the quality of feature selection, and the medical characteristics of ECG depend on whether the signal feature points are accurate. Errors in locating points can be amplified during diagnosis. Secondly, capturing subtle morphological differences in different types of arrhythmias is challenging due to the heart's complexity and the population's specificity. Thus, these limitations restrict the model's performance, leading to a high misdiagnosis rate in practical application[9]. Recently, deep learning (DL) technologies have made great progress in areas like computer vision and natural language processing. The ability to apply DL methods reasonably and effectively to arrhythmia diagnosis has attracted the attention of numerous researchers, which further provides fascinating insights into remote ECG automatic diagnosis[10–14]. A DL-based model is to obtain input-output mapping function by learning hidden rules in data. In imaging diagnosis, DL-based models achieve better performance than human experts, such as cancer diagnosis[15] and CT scanning[16]. Although training DL-based models can achieve efficient performance, it requires a large amount of labeled data, which brings some challenges for medical applications[17]. Thus, the application of DL to arrhythmia diagnosis requires there to be reliable arrhythmia databases that contain high-quality and well-labeled ECG recordings[9]. In recent years, the Massachusetts Institute of Technology Beth Israel Hospital arrhythmia (MIT-BIH) database[18] and the American Heart Association (AHA) dataset[19] have been widely used by researchers. However, due to the limitations of a small number of patients, limited arrhythmia types, and lack of multi-label ECG recordings in the dataset, the DL-based model cannot learn appropriate representations from these open datasets. At the start of our project, we were concerned that other largescale datasets like CODE[9,20] and PTB-XL[21] would not be suitable for our study as they do not contain data collected from the Chinese population or the restricted disease category in the CODE[9,20] dataset that excludes atrial premature contraction (PAC) and ventricular premature contraction (PVC).

In remote and underdeveloped areas, the accuracy and efficiency of diagnosing arrhythmias through Electrocardiogram (ECG) recordings can vary greatly among clinicians. To address this, we explored the potential of using digital ECG and artificial intelligence (AI) technology to assist in diagnosing arrhythmias outside of hospitals. Over the course of more than a decade[22–25], our team has diligently worked towards the development of a wearable ECG acquisition device, accompanied by a mobile application with a cloud-based AI diagnosis model, to provide an AI-aided solution for out-of-hospital ECG diagnosis in remote and underdeveloped areas. We collected a large dataset of ECG recordings from over 270,000 patients in China, covering common arrhythmias. Experienced cardiologists labeled the dataset. Using deep learning, we developed a model that accurately diagnoses multiple arrhythmia types. Our model achieves performance comparable to that of cardiologists in terms of F1 score, AUC-ROC, and accuracy. Additionally, our model provides explainable insights into the diagnostic process, allowing clinicians to quickly identify abnormal regions on the ECG and improve the accuracy and efficiency of ECG diagnosis. Overall, our diagnostic system shows promising results in improving out-of-hospital ECG diagnosis, which can enhance telemedicine and productivity in clinical settings.

Considering the application scenario of telemedicine, single-lead ECG acquisition is convenient and more suitable for out-of-hospital ECG diagnoses in remote and underdeveloped areas. This paper describes a new large-scale, well-labeled clinical multi-label single-lead dataset consisting of various arrhythmias in China. The dataset contains 352,725 clinical recordings ranging from 10-45 s in duration and sourced from 272,753 different Chinese patients between January 2017 and December 2021. We also describe a deep learning-based multi-label interpretable diagnostic model that can diagnose 6 common arrhythmias from ECG readings.

## Methods

The main challenge in implementing an automatic ECG diagnosis system in practice is the absence of a standardized and comprehensive ECG dataset. Nowadays, ECG recordings are mainly stored in image or PDF formats, which are unsuitable for training AI models. Furthermore, open-source ECG datasets collect few patients, cover fewer types of arrhythmias, and have no complete diagnostic notes. Thus, with the assistance of Shanghai First People's Hospital, we constructed a large-scale Chinese ECG dataset (LSCP-ECGDS) that could meet the requirements of out-of-hospital ECG diagnosis.

And then, we utilize a deep neural network (DNN) along with medical diagnosis knowledge for building an accurate arrhythmia classification model (SJTU-ECGNet). Figure 1 illustrates the design of the SJTU-ECGNet arrhythmia classification model. Meanwhile, the diagnostic system can provide clinicians with direct evidence for review and visual inspection by simultaneously outputting the prediction results and the model's attention on each region of the input ECG, revealing the decision-making reasoning process of the model. As a result, it helps to reduce the variability of clinical decision-making. These neural networks are interdependent and complementary in reaching the final diagnosis.

### Dataset sources

As shown in Supplementary Fig. 1a, the 12-lead ECG recording contains 3 pressurized limb leads, 3 bipolar limb leads, and 6 chest leads. To ensure the feasibility of remote real-time diagnosis, the wearing convenience of sampling equipment needs to be considered. Thus, we describe the long-term monitoring of lead II ECG, which can not only ensure the preliminary diagnosis of common arrhythmia diseases but also ensure the convenience of wearing the sampling equipment. Additionally, single-lead ECG wearable devices offer convenience for large-scale out-of-hospital cardiovascular diagnosis in remote and underdeveloped areas of China. To obtain a high-quality model despite variations in length and noise interference of collected ECG recordings, Supplementary Fig. 1b shows that 51,261 high-quality ECG recordings are carefully selected and divided into four parts, including training set, verification set, test set, and hidden set. Specifically, the training set is applied in the modeling process to obtain useful model weights. Then, the verification set is used for fine-tuning the model parameters and structure, and to gauge the presence of over-fitting. Further, the test set is used to evaluate the model's performance, such as accuracy. Additionally, the hidden set is comprised of 1261 ECG recordings that are independently labeled by three clinical cardiologists. The hidden set is used to compare the diagnostic performance differences between our SJTU-ECGNet model and three clinical cardiologists, which results in evaluating the effectiveness of our model in clinical screening.

We train SJTU-ECGNet to detect Normal sinus rhythm (Normal), sinus bradycardia (SB), sinus tachycardia (ST), atrial premature contraction (PAC), atrial fibrillation (AF), and ventricular premature contraction (PVC). The six types are shown in Supplementary Fig. 1c and can be chosen as large-scale screening indicators for cardiovascular diseases, thus providing early warning for the sub-health status of the heart and reducing the risk of sudden cardiac death. For instance, AF is often asymptomatic, making a diagnosis difficult without screening, and is recognized only when associated with complications[26]. For patients with PVCs, the risk of sudden death is higher if PVCs occur frequently after myocardial infarction (MI)[27].

A total of 51,261 lead II ECGs were collected for model training, validation, and hidden testing. Among these, 40,000 ECGs were allocated to the training dataset, 5,000 ECGs to the validation dataset, another 5,000 ECGs to the validation dataset, and 1261 ECGs to the hidden dataset. Initially, 352,725 clinical 12-lead ECGs were obtained from 272,753 patients. 29,315 ECGs were excluded due to poor signal quality. Additionally, 94,322 ECGs were excluded as they contained not only the six diseases of interest but also other types of arrhythmias. Furthermore, 46,366 ECGs were excluded to eliminate duplicates, and 131,461 ECGs were selected as the external test set for evaluating the model generalizability of the proposed SJTU-ECGNet structure. MIT-BIH[18], CODE[9,20], and PTB-XL[21] datasets were also used to verify generalization, following the terms

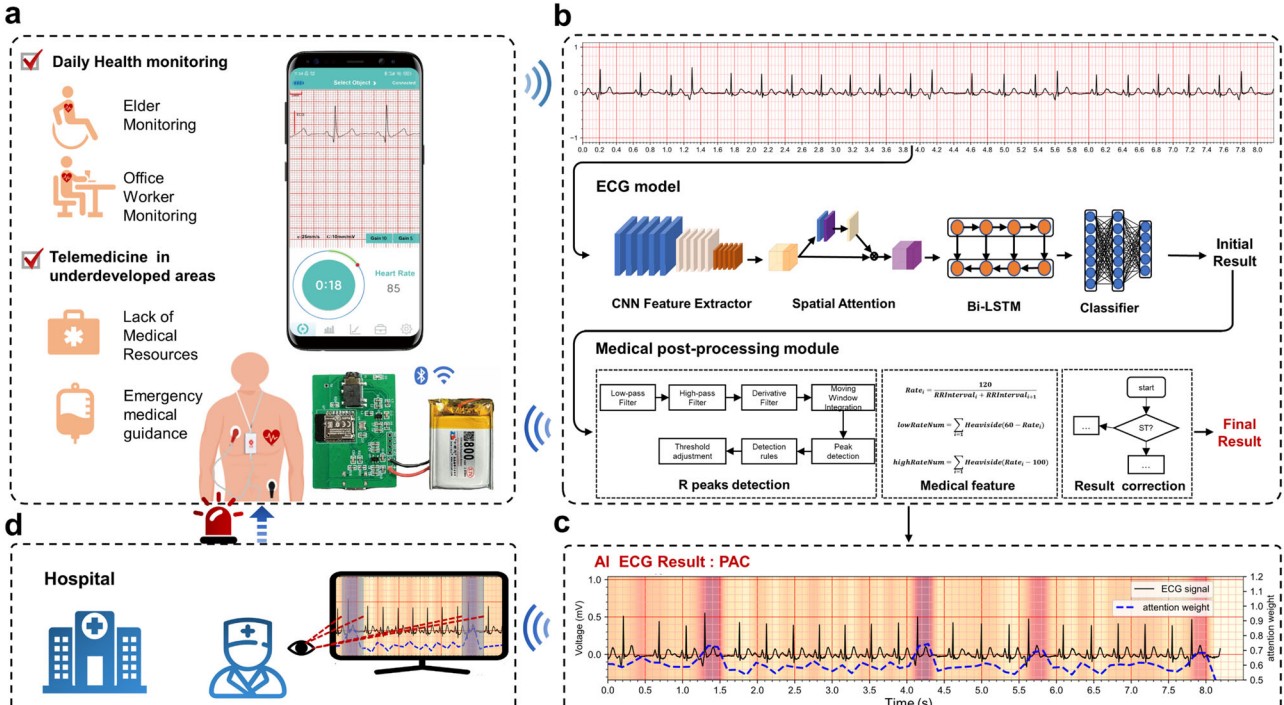

**Fig. 1 | Diagnostic system framework. a** Demand and application for arrhythmia telemedicine. Considering the daily health monitoring and remote cardiovascular disease diagnosis in underdeveloped areas, a reliable and accurate automatic arrhythmia diagnosis system is urgently needed. **b** The model structure of the diagnostic system, including a DL-based Module with several DNNs to output initial results and a Medical Post-processing Module to obtain the final results. **c** The AI ECG result includes the interpretability of the model. Corresponding attention weights are represented as attention maps, illustrating significant regions the network pays more attention to when describing ECG signals. **d** The generated attention maps reveal the decision-making mechanism of clinicians, which facilitates the 24/7 assessment of people's cardiac health status without leaving home.

stipulated at their websites https://physionet.org/content/mitdb/1.0.0/, https://zenodo.org/record/4916206#.YcQzSmBBwdU,https://physionet.org/content/ptb-xl/1.0.1/. (Results and Discussions are shown in the Supplementary Discussion Section.) A comprehensive description of the screening process is provided in Supplementary Fig. 1(d).

### Arrhythmias classification Model and method overview

The arrhythmias classification studied in this paper belongs to finite-length time-series signals classification[28]. Arrhythmias are commonly classified using ECG recordings that are of varying lengths due to the sampling environment of clinical ECG. Thus, models that are capable of handling varied-length input are necessary. Moreover, when an arrhythmia occurs, local and global ECG features are both required for an accurate diagnostic result. Hence, the model structure should meet the requirement of capturing imperative information.

Supplementary Fig. 2 shows the schematic diagram of the proposed DL-based and medical-knowledge-involved arrhythmias classification method, which can achieve complementary advantages, aiming to balance the data dependency[29] of the model with the influence of medical definition. The details of the proposed structure are shown in Supplementary Methods. Data preprocessing is listed in Supplementary Fig. 3. The whole classification method comprises these two modules, and the overview of each module is listed as follows.

For the DL-based Module: There are mainly three stages in this module: (i) Data reprocessing, (ii) Feature representation, and (iii) Multi-label output. To extract high-dimensional feature representation in stage (ii), we design a deep neural network integrating 1-dimensional convolution layers, residual blocks, BiLSTM layer, and ATT layer.

For the Medical Post-processing Module: After obtaining the initial label from the former module, which contains specific arrhythmias, we adopt an automated medical knowledge method to perform the post-processing of the results.

### Neural network architecture

Although the change in ECG is minimal, even subtle changes can have a noteworthy impact on the result. We design CNNs, Residual blocks, and BiLSTM layer for automatically extracting features from ECGs, as shown in Supplementary Fig. 4. The Residual block is responsible for extracting the short-term dependence features, while the BiLSTM layer extracts the long-term dependence features of ECG. Then, the ATT layer is utilized to enhance beneficial features and indicates the model's focus on the input features. Finally, the multi-label classifier processes the deep features to generate the prediction results. Following the DL-based prediction results, the medical post-processing module corrects some arrhythmia disease prediction results, including ST and SB. Combining a DL-based model and medical post-processing module, we use the ECG characteristics of different arrhythmias to improve the final diagnostic accuracy. Besides, the details of the arrhythmias diagnosis model are listed in Supplementary Figs. 5–8, and Supplementary Table 1.

### Data relabeling process

To evaluate the clinical performance of our model, we constructed a representative hidden dataset and invited three experienced cardiologists to annotate this dataset independently. The cardiologists did not discuss or communicate during this diagnosis process.

### Statistical analysis

To comprehensively evaluate the formidability and correctness of the model, we use accuracy (Acc), macro F1 score (F1-macro), and the area under the curve of the receiver operating characteristic (AUC-ROC) as performance metrics[22]. The F1-macro evaluates the overall performance of the model and is robust to data imbalance effects. The detailed formulas can be found in the Performance metric Section of the Supplementary Methods.

The confidence interval (CI) was derived from bootstrap distribution, that is, to estimate parameters using a repeated sampling method. The 95%

CI can be expressed as Eq. (1).

$$(\hat{\theta} - t_{b,0.025}^* \times \hat{\sigma}_{\hat{\theta}}, \hat{\theta} + t_{b,0.025}^* \times \hat{\sigma}_{\hat{\theta}}) \quad (1)$$

## Implementation details

We utilize NVIDIA GTX 1080 GPU to train the proposed AI-aided model and apply Adam optimizer[30] to update network weights during the back-propagation process. The learning rate is set to 0.001. Then, during the training process, Focal Loss is selected as a loss function with parameter $\gamma = 2$. The model weights are initialized with kaiming uniform. The batch size is set as 128. The whole training process runs for 50 epochs, and the final model is selected based on the best validation results obtained during the optimization process.

## Ethics statement

The methods were performed in accordance with relevant guidelines and regulations and approved by Shanghai Jiao Tong University and Shanghai First People's Hospital Affiliated to Shanghai Jiao Tong University Review Board, protocol 2018KY155-2. Due to the large scale and anonymization of the data used in this study, the Review Board waived the requirement for informed consent from each patient.

## Reporting summary

Further information on research design is available in the Nature Portfolio Reporting Summary linked to this article.

## Results

To evaluate the effectiveness of SJTU-ECGNet, the following ablation experiments and comparison experiments were designed, which were listed in Supplementary Fig. 9. Besides, the other hyper-parameters of baseline models were the same as SJTU-ECGNet.

## Ablation Experimental results of the model

The ablation experimental results present that the BiLSTM layer extracts the long-term correlated features of ECG signals. The manifestation of arrhythmia on ECG is not only the change of waveform but also the variations of rhythm between different QRS waves. The BiLSTM layer effectively focuses on the changes in ECG rhythm, which improves the diagnostic performance. Besides, the combination of CNNs, residual blocks, BiLSTM layer, and ATT can effectively enhance diagnostic performance by extracting more meaningful features. Further, compared with the proposed method without the medical post-processing module, SJTU-ECGNet can effectively correct the diagnostic errors of DNN in sinus arrhythmias, and effectively complement the DL-based model. Generally, the waveform of sinus arrhythmia disease is normal, but its rhythm and RR interval change are abnormal. These are clear medical judgment criteria in clinical medical diagnosis. Therefore, medical knowledge can better rectify the deficiency of the DL-based model in the diagnosis of this kind of arrhythmia and then effectively improve the final diagnosis accuracy. Then, we replicated the ResNet-based model (Model e) and VGG model (Model f) from previous studies on our dataset. The results in Tables 1 and 2 indicate that the ResNet-based model achieves an accuracy of 86.68% on the test dataset and 73.83% on the Hidden dataset. In addition, we implemented the VGG model (Model f) to classify ECG in the same configuration, achieving 89.60% and 76.84% accuracy on the test and hidden datasets, respectively. Hence, the proposed SJTU-ECGNet achieves better accuracy on both the test and hidden datasets compared to these standard neural network models. Generally, the proposed model architecture is reasonable and has a good diagnostic performance based on the conducted ablation experiment. What's more, we validated the performance of the model on three additional publicly available datasets (MIT-BIH[18], CODE[9,20], PTB-XL[21]), compared the post-processing effects under different R peaks location algorithms, and evaluated the computational speed of the model. Detailed information is listed in Supplementary Tables 2–5, and Supplementary Fig. 10.

## Model performance on comparison experiments

Table 1 and Table 2 list the comparison results between SJTU-ECGNet and baseline models in the test set and hidden set. Compared with baseline models, the proposed method achieves better and more stable model performance, with F1-macro and accuracy scores of 85.04% and 95.50%, respectively, in the test set and 83.51% F1-macro and 93.74% accuracy in the hidden set. According to the experimental result, the average accuracy of the

**Table 1 | Comparison results of baseline models and SJTU-ECGNet in the test set**

|  | a | b | c | d | e | f | **ECGNet** |
|---|---|---|---|---|---|---|---|
| *F1* |  |  |  |  |  |  |  |
| Normal | 93.63 | 94.55 | 94.45 | 94.78 | 92.59 | 94.21 | **97.74** |
| ST | 69.80 | 68.20 | 64.95 | 67.86 | 59.67 | 68.56 | **90.69** |
| SB | 63.37 | 71.47 | 68.71 | 65.86 | 61.60 | 69.58 | **89.77** |
| PAC | 73.29 | 69.88 | 69.94 | 69.46 | 36.36 | 64.20 | **69.46** |
| AF | 79.66 | 87.10 | 87.02 | 90.77 | 60.94 | 84.85 | **90.77** |
| PVC | 73.87 | 77.88 | 81.08 | 71.84 | 61.05 | 74.77 | **71.84** |
| *Average* |  |  |  |  |  |  |  |
| Accuracy | 88.52 | 90.36 | 89.98 | 90.42 | 86.68 | 89.60 | **95.50** |
| F1 | 75.60 | 78.18 | 77.69 | 76.76 | 62.04 | 76.03 | **85.04** |

The bold values represent the best scores.

**Table 2 | Comparison results of baseline models and SJTU-ECGNet in the hidden set**

|  | a | b | c | d | e | f | **ECGNet** |  |
|---|---|---|---|---|---|---|---|---|
| *F1* |  |  |  |  |  |  | F1 (95% CI) | AUC ROC (95% CI) |
| Normal | 86.13 | 86.38 | 86.73 | 86.62 | 84.88 | 86.28 | **97.32(96.67-98.04)** | 0.934(0.916–0.955) |
| ST | 67.76 | 59.04 | 61.54 | 53.42 | 60.26 | 61.15 | **94.18(90.86-96.60)** | 0.995(0.991–0.998) |
| SB | 57.14 | 59.74 | 68.87 | 63.44 | 48.37 | 64.55 | **85.00(80.00-89.61)** | 0.971(0.947–0.990) |
| PAC | 61.90 | 51.28 | 60.00 | 56.41 | 11.43 | 48.78 | **56.41(40.49-73.78)** | 0.965(0.929–0.986) |
| AF | 78.95 | 90.48 | 87.80 | 82.93 | 52.63 | 81.08 | **82.93(72.69-93.02)** | 0.998(0.995–1.000) |
| PVC | 81.97 | 88.52 | 90.32 | 85.25 | 67.86 | 84.75 | **85.25(71.77-93.06)** | 0.997(0.992–0.999) |
| *Average* |  |  |  |  |  |  |  | *AUC ROC* |
| Accuracy | 76.61 | 76.92 | 78.27 | 76.92 | 73.83 | 76.84 | **93.74** | 0.977 |
| F1 | 72.31 | 72.57 | 75.88 | 71.35 | 54.24 | 71.10 | **83.51** |  |

*AUC ROC* area under the receiver operating characteristics curve, *CI* confidence interval. The bold values represent the best scores.

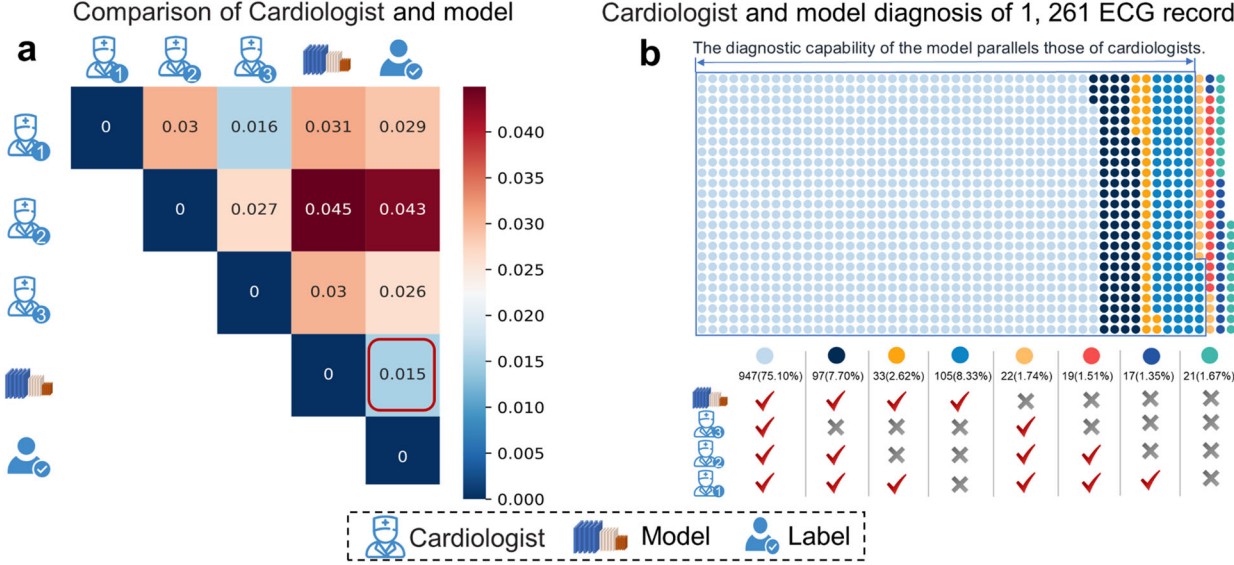

**Fig. 2 | Comparison of SJTU-ECGNet and cardiologists in diagnosis effect.**
**a** Diagnostic inconsistency matrix between cardiologists and SJTU-ECGNet. (As the value decreases, the diagnostic consistency between the two increases. The smallest value is marked with a red box.) **b** Diagnostic correctness charts in each ECG recording (total of 1261 ECG recordings). The dots in the polygonal region indicate that the performance of the model is comparable to, or even higher than, that of the cardiologists. The dots outside the region represent correct diagnoses made by cardiologists but missed diagnoses or misdiagnoses by the model. For instance, the light blue dots indicate that 947 recordings are accurately diagnosed by all three cardiologists and the SJTU-ECG model, and the black dots represent 97 recordings that are correctly diagnosed by the model and two cardiologists but are misdiagnosed by the third cardiologist, and so on. The red tick mark denotes the correctly classified recordings, meaning that the predicted label set is identical to the ground-truth label set. The black cross mark indicates that the predicted label set is either partially or entirely incorrect compared to the ground-truth label set.

three cardiologists is 84.30%, while the diagnostic system achieves an average accuracy of 93.74%, which exceeds that of cardiologists. A comparison of three cardiologists and SJTU-ECGNet in diagnostic accuracy and F1-macro in the re-label hidden set can be found in Supplementary Fig. 11.

Moreover, the mean AUC ROC score of all 6 classes is 0.977 in the hidden set. Combining with Supplementary Fig. 12, we obtain the AUC ROC score for each class, where the lowest AUC score is 0.934 (0.916–0.955, 95% CI) for the Normal class and the highest score is 0.998 (0.995–1.000, 95% CI) for Atrial Fibrillation class.

Additionally, comparing the diagnostic results of the model with the committee consensus label, confusion matrices of the test set and hidden set are shown in Supplementary Fig. 13, which shows the absolute number of (i) false positives, (ii) false negatives, (iii) true positives; (iv) true negatives, for each abnormality on the dataset.

### Comparison performance of the model with cardiologists

To further compare the diagnostic effectiveness of SJTU-ECGNet with that of cardiologists, the diagnostic inconsistency matrix and diagnostic correctness charts are respectively proposed in Fig. 2a, b. We use hamming loss to calculate the diagnostic inconsistency between the model and cardiologists (The detailed calculations can be found in Supplementary Results). There exist differences in the diagnostic inconsistency between different cardiologists, the inconsistency of cardiologist1 and cardiologist3 is 0.016, the inconsistency of cardiologist1 and cardiologist2 is 0.03, and the inconsistency of cardiologist2 and cardiologist3 is 0.027. Compared to the ground-truth label, the inconsistency of three cardiologists is respectively 0.029, 0.043, and 0.026, while the inconsistency of SJTU-ECGNet is 0.015, much smaller than that of cardiologists, which reflects better consistency than cardiologists. Among the 1261 ECG recordings, 947 (75.10%) are accurately diagnosed by three cardiologists in accordance with the SJTU-ECGNet model, while 97 (7.70%) are misdiagnosed by the third cardiologist. Additionally, 33 (2.62%) recordings are accurately diagnosed by a single cardiologist and the SJTU-ECGNet model, and 105 (8.33%) recordings are solely diagnosed correctly by the SJTU-ECGNet model. There also exist recordings that are ignored by the model while diagnosed correctly by

cardiologists, that is, 22 (1.74%) recordings for three cardiologists, 19(1.51%) recordings for two cardiologists, and 17 (1.35%) recordings by one cardiologist. Additionally, three cardiologists and the model all get wrong diagnoses in 21 recordings.

Previous research suggests that 80% accuracy is achievable for diagnosing common arrhythmias automatically. Our approach outperforms this benchmark by achieving over 93% accuracy on various datasets, as shown in Tables 1 and 2. Meanwhile, an analysis of the model's performance in classifying individual types of arrhythmias reveals an average F1 score of over 83% and a score of over 82% for serious and common arrhythmias like AF and PVC, indicating that the model has an effective diagnosis ability for arrhythmias despite the unbalanced dataset. Furthermore, the medical post-process module substantially improves the F1 score of sinus rhythm arrhythmias, including Normal, ST, and SB. Table 1 reflects that in the test set, the F1 score of Normal increased from 94.78 to 97.74, for ST, it increased from 67.86 to 90.69, and for ST, it increased from 65.86 to 89.77. The experimental results highlight the effectiveness of the medical post-process module in enhancing the performance of detecting sinus rhythm arrhythmias. Here, we also study the differences in ECG diagnosis among cardiologists and further compare the results with our approach. We assess cardiologist differences in diagnosis using 1261 ECG recordings, as shown in Fig. 2b. Despite similar academic backgrounds or institutional affiliations, the diagnosis results on the hidden set vary considerably among cardiologists. The results show that more than 173 ECG recordings exhibit inconsistent diagnoses by different cardiologists, accounting for about 14%.

### Discussion

The diagnosis of abnormal rhythm or beat is an important technique to realize early and timely warning. While maintaining convenience, single-lead ECG can provide valuable information on common arrhythmias to ensure effective out-of-hospital screening. Nevertheless, given that ECG changes can be subtle and associated with multiple diseases, cardiologists and traditional automatic diagnostic systems face various challenges in accurate arrhythmia diagnosis. The proposed DL-based model is trained with a large-scale clinical ECG dataset, which can accurately diagnose multi-

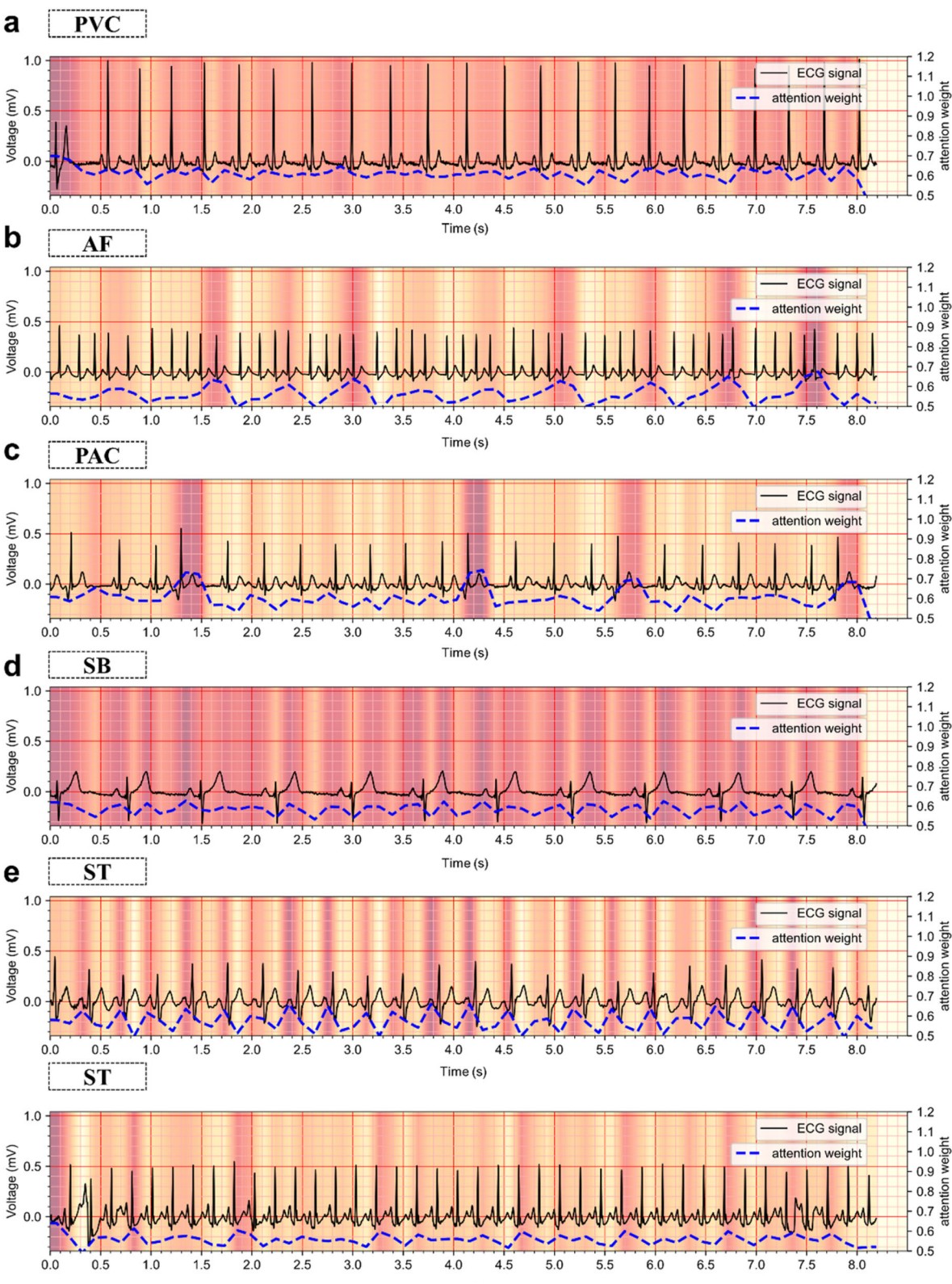

**Fig. 3 | The attention map.** Corresponding attention weights are represented as attention maps for PVC (**a**), AF (**b**), PAC (**c**), SB (**d**), and ST (**e**), indicating useful regional features the network pays more attention to when describing ECG signals. **a** The ECG presents ventricular premature beat, which is located at 0–0.5 s. The attention map reaches a peak and shows the PVC area clearly. **b** The ECG shows persistent atrial fibrillation. The attention map reaches several peaks non-periodically, which elaborates the irregularity of AF ECG. **c** The ECG presents four atrial premature beats. The attention map reaches four peaks, respectively. The locations of peaks correspond to the abnormal beats. **d**, **e** Sinus bradycardia and sinus tachycardia ECG have more stable attention values. The attention map reaches several peaks periodically, and peak values are smaller. The peak value of ST is larger and more intensive than the SB's.

label arrhythmias and ensure the model's interpretability. The above-mentioned detection performance of the diagnostic system is stable. Our model achieves an impressive performance with 93.74% accuracy, surpassing that of cardiologists. The results show that this diagnostic system has the potential to be applied in early clinical diagnosis and can further relieve the situation of the shortage of medical resources.

Next, we demonstrate qualitatively the ECG diagnostic process of our method. The results of ECG diagnosis and the degree of attention paid by the model to each region (attention map) are shown in Fig. 3. Among them, the attention map represents the input signal regions useful for model diagnosis in the arrhythmia detection process, identifying the regional features that the network prioritizes the most when describing ECG signals. Taking PAC as an example, the model mainly focuses on the early occurrence of waveforms in a specific segment of the ECG signal, consistent with the clinician's diagnostic criteria. By extension, we analyze the attention maps generated by the model when diagnosing different types of abnormal ECG signals and observe that the attention maps are congruent with the clinician's medical interpretation. This indicates that our method can accurately locate the effective regions of ECG signals and then enhance the characteristics of these regions to ensure the final precise prediction.

Finally, we present the feasibility of ECG end-to-end diagnosis, representing a substantial departure from conventional ECG automatic diagnosis methods. Many traditional approaches, such as the University of Glasgow's ECG analysis program[31], rely on traditional signal processing approaches to extract various features and then design corresponding feature classifiers to obtain the final classification results. The end-to-end automatic diagnosis simplifies the diagnosis process, avoids the cumulative error caused by the feature extraction process, and can achieve better performance in the clinical environment[9]. However, end-to-end ECG diagnosis requires a large amount of high-quality labeled data, which puts forward requirements on the size of the dataset. On the one hand, several existing methods are based on the MIT arrhythmia dataset[18] for model training, but these databases do not have a considerable number of patients, which restricts meeting the data scale requirements of end-to-end model training. On the other hand, some previous works have obtained datasets of a certain size from 12 leads ECGs or hand-held devices, such as CODE[9,20], CinC Challenge 2017[32], and PTB-XL[21], but these datasets either do not contain multiple arrhythmias coupling conditions[10] or lack disease categories of interest. Consequently, they do not fulfill the requirements for an end-to-end diagnostic model for clinical needs. Our dataset consists of short ECG recordings obtained from clinical examinations, which includes not only arrhythmias, such as SB, ST, and AF, as described in previous research[9], but various types of common premature beats, such as PAC and PVC. Unlike the MIT-BIH[18] database, where each beat corresponds to one label, our dataset provides long-term ECG signals with clinically diagnosed conclusions that adhere to clinical practices. Based on the high-quality LSCP-ECGDS constructed here, it enables the expansion of preliminary findings of DL-based models in automated ECG diagnosis to various clinical diagnostic applications. Using the increasingly available ECG data, this technological development can lead to high-precision ECG diagnostic systems, which can minimize inaccuracy in diagnoses while providing access to timely diagnosis of various cardiovascular diseases.

In summary, we construct a large-scale Chinese ECG benchmark dataset from 272,753 patients. In addition, we propose a DL-based arrhythmia diagnosis model combined with medical knowledge. The proposed SJTU-ECGNet enables end-to-end clinical diagnosis of ECGs, thereby potentially revolutionizing healthcare in remote and under-developed regions. In these areas, cardiovascular disease is the leading cause of death[33]. For 5 common arrhythmias and sinus rhythms, the results demonstrate that the model achieves an F1-macro score of 83.51% and accuracy of 93.74% and has superior diagnostic performance compared to that of cardiologists. Meanwhile, the interpretable qualitative analysis of the prediction results indicates that the diagnostic mechanism of the model is similar to the clinician's criterion to a certain extent. We explore the potential of utilizing automated AI-aided ECG diagnosis to streamline screening processes and alleviate the workload of clinicians. The proposed diagnostic system described here could play an important role as a tool used by cardiologists to help control cardiovascular diseases. With accurate dynamic monitoring of cardiovascular health, people could receive 24/7 assessment of their cardiac health status without leaving home, thus changing the conventional interactions required between the doctor and patient during ECG monitoring for the detection of arrhythmias.

## Data availability
The hidden test set used in this study is openly available and can be downloaded at https://github.com/xin-gou/automatic-ecgdiagnosis and https://zenodo.org/records/10177838[34]. Requests to access the training data will be considered on an individual basis by the Department of cardiology, Shanghai First People's Hospital, Affiliated with Shanghai Jiao Tong University. Any data use will be restricted to non-commercial research purposes, and the data will only be made available on the execution of appropriate data use agreements. The source data underlying Figs. 2 and 3 are provided as a Source Data file in Supplementary Data 1. The external hospital set comes from the Shanghai First People's Hospital, Affiliated with Shanghai Jiao Tong University, with the same acquisition access as the training set. The MIT-BIH[18] dataset is available at: https://physionet.org/content/mitdb/1.0.0/. The CODE[9,20] dataset is available at: https://zenodo.org/record/4916206#.YcQzSmBBwdU. The PTB-XL dataset[21] is available at https://physionet.org/content/ptb-xl/1.0.1/.

## Code availability
The code for using the proposed SJTU-ECGNet is openly available at Github repository: https://github.com/xin-gou/automatic-ecgdiagnosis. The trained weights of the proposed model is available in https://zenodo.org/records/10176213[35].

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

## Acknowledgements

This paper is partly supported by the National Key R&D Program of China (No. 2018YFB1307005), National Postdoctoral Program for Innovative Talents (No. BX20230215), China Postdoctoral Science Foundation (No. 2023M732219), Shanghai Municipal Science and Technology Major Project (No. 2021SHZDZX0102) and Shanghai Municipal Health and Family Planning Commission smart medical key project (No. 2018ZHYL0226).

## Author contributions

Y.J. conceived the study, designed the experiments, and conducted all experiments. Y.J., Z.L. and M.W. wrote codes for utilizing knowledge-fused DL-based methods for arrhythmia detection. Z.L. and J.L. pre-processed the input ECG data. Y.T., Y.L., X.W. and C.L. gave advice on writing and experiment design. L.Z. manually audited the ECG recordings and results for the cardiologist label. Y.J., Z.L. and M.W. wrote the manuscript. All authors critically revised the manuscript.

## Competing interests

The authors declare no competing interests.
