## [Peer Review File · Communications Medicine]

Reviewers' comments:

Reviewer #1 (Remarks to the Author):

This is a review of the paper by Jin et al., on a cardiologist-level knowledge-fused Deep Learning model for arrhythmia diagnosis. The authors justify the need for such a model to help with automated electrocardiogram (ECG) readings, especially in the remote and underserved areas of China. The authors used data from January 2017 to December 2021, and studied to 272,753 Chinese patients. They achieved a 93.74% accuracy and reported that the diagnosis of 6 common arrhythmias using the model was better than performance of cardiologist. What is commendable is that the authors built a large clinical dataset consisting of single-lead ECG recordings of 10-45 seconds duration. The term the model as SJTU-ECGNet which was developed as a combination of a deep learning neural network (DNN), and medical diagnosis knowledge. To train the model 51,261 ECG recordings were used. The whole dataset was divided into 4 parts, including 40,000 ECGs for training set, 5,000 ECGs for verification set, 5,000 ECGs for test set and 1,261 ECGs for hidden set (cardiologist testing).

These are several concerns that exist:

Major:

- The introduction section is quite lengthy and stretched out. There are details that are already mentioned in the methods section so these can be cut short.
- In data pre-processing, the authors describe undersampling to normalize all ECG recordings to a standard of 4096 sample points. It seems that this changes the width of some of the complexes. What are the challenges if model training is performed on raw samples with their length differences? Was model performance made better by the undersampling, or was that done to make it more intuitive to humans?
- The authors used a neural network architecture consisting of 1D convolutional neural network (CNN), residual blocks, bidirectional long short-term memory (BiLSTM layer), and attention maps (ATT) for model explainability. Did the authors try a standard neural network model, before attempting more complex architecture?
- What was the total computational time needed to train the ECGs for this study?
- There was a medical post-processing module after the initial neural network training. The authors mention the need for careful heart rate measurement, peak detection for RR intervals as a reason for this. These tasks have been shown to be possible with neural networks. Examples include peak prominence in Python or the various filters available in Matlab.
- It seems cumbersome to include a post-processing module with clinician's reading thousands of ECGs. The question that arises is whether this step is even required? Why not trust the model to perform without the clinician post-processing? It appears, based on Table 1 that Model d that was without post-processing had the same AUROC as the SJTU-ECGNet in prediction of PAC, AF, and PVC. So, is it justified to have clinicians go over thousands of ECGs to help differentiate sinus tachycardia from sinus bradycardia?
- The training data was exclusively made up of Chinese patients. How do the authors feel about clinical use and generalizability in centers across the world? Unless the model is validated in other datasets, this is an expected but significant issue that must be mentioned in the limitations section.

Minor:

- Did the authors try and analyze the ECG recordings of the 24.9% of patients that had their ECG misinterpreted by at least one of the three cardiologists, but with better performance by the model?
- Multiple spelling and grammatical errors throughout the paper.

Reviewer #2 (Remarks to the Author):

I congratulate the authors for putting in great efforts to make automated ECG diagnosis of the common arrhythmia, which may go a long way into helping patients in remote areas. I have a few minor comments.

Comment 1

The introduction can be rephrased to exclude details of the methods like – “This paper builds a well-labelled, large-scale clinical multi-label single-lead dataset with different arrhythmias of China for model training and evaluating, containing 352, 725 clinical recordings with 10-45s duration from 272, 753 different Chinese patients from January 2017 to December 2021. Then, we design the classification arrhythmia methods (SJTU-ECGNet) by combining deep neural network (DNN) and medical diagnosis knowledge.....”

Comment 2

The description of the conceptual model of the diagnostic system is very neatly described but it would be great to include it in the methods section rather than introduction

Comment 3

The figures should be included in the chronological order in the manuscript e.g. Supplementary Figure 8 will come into the description after 1. Please change the numbers of figures wherever needed

Comment 4

“The important reason why the automatic diagnosis system is difficult to be applied in practice is the lack of a complete and standardized ECG dataset. Nowadays, ECG recordings are mainly stored in image or PDF format, which is difficult to be applied in training AI models. And then, the open-source ECG datasets collect few patients, cover fewer types of arrhythmias, and have no complete diagnostic notes. Therefore, with the assist of Shanghai First People's Hospital, this paper constructs a large-scale Chinese ECG dataset (LSCP-ECGDS) for meeting the requirements of out-of-hospital ECG diagnosis. Further, we design different experiments for validating the feasibility of our approach.” This doesn't need to be in the results section. The authors may consider including it in the methods under a separate heading may be

Comment 5

Kindly reword this statement with less redundancy – “For 1261 ECG recordings, 947(75.10%) recordings are all diagnosed correctly by three cardiologists and SJTU-ECG model, 97(7.70%) recordings are diagnosed correctly by two cardiologists and SJTU-ECG model, 33(2.62%) recordings are diagnosed correctly by one cardiologist and SJTU-ECG model and 105(8.33%) recordings are only diagnosed correctly by SJTU-ECG model.”

Comment 6

Discussion starts off with a good content but then the next 2 paragraphs again try to describe the results. It will be good to remove the duplication of wordings in the discussion section as compared to results and focus on the discussion on the salient features in the study per se so that the reader will understand the highlights of the study.

Comment 7

The paper needs proofreading to avoid incomplete statements like - "This paper describes the potential of automated AI-aided ECG diagnosis, which will bring greater convenience to clinical practice by fully developing."

Comment 8

There are minor grammatical errors which may be helped by proofreading.

Reviewer #3 (Remarks to the Author):

The authors provide a new dataset and develop a model for classifying 6 types of arrhythmias. I believe the general contribution is interesting, but I believe the manuscript is unclear in some key points.

There are also parts of the text that seem to be contradictory.

- There seems to be some contradicting information: In the introduction, it is said that the dataset is a single lead, and this is the impression that I got from Figure 1. But in Supplementary Figure 8 it says that it is a 12-lead recording (and also in the method section).
- The data exclusion process is unclear. In the introduction and abstract, it mentions 272,753 Chinese patients. But in methods, it says that 51,261 ECGs are used in the development. Maybe adding a diagram in the style of STARD diagram to report the flow of participants through the study could fix the problem.
- The dataset is not properly described. Supplementary table 8 does some description. But the absolute numbers are absent. I believe this is a key piece of information that is missing. It is also unclear to be how many exams in each class there are. I believe a table would be significantly more transparent.
- It is unclear from the text why the author divides into two different test sets. What is exactly the role of the Hidden test set? In Supplementary table 8, it says that it is for Doctor testing. But it is not clear to me why the division.
- Why not compute the F1 score and accuracy of the doctors?
- I think it would be interesting if the authors also provided a table with other traditional metrics (Specificity, sensitivity, PPV, NPV)
- I found Figure 2 quite unclear. It is hard to read the results from it.
- It is not clear how the threshold was selected. It is also unclear to me if a softmax or a sigmoid is being used (Sup. Figure 6 and 9 seem to contradict each other).

Some minor points:

- In the Methods description, too much space is devoted to describing well-known performance metrics. And the description of the implementation details is very short.
- In the discussion, the authors mention only a small size open-source ECG dataset. But there are some large-scale open-source datasets. I mention PTB-XL(<https://doi.org/10.1038/s41597-020-0495-6>), CODE (<https://doi.org/10.17044/scilifelab.15169716> - available for research upon request) and CODE 15% (<https://doi.org/10.5281/zenodo.4916206>)

Summary of Responses and Amendments

Manuscript ID: **COMMSMED-23-0107-T**

Manuscript Title: **Cardiologist-level interpretable knowledge-fused deep neural network for automatic arrhythmia diagnosis**

Journal: **Communications Medicine**

Dear Reviewers,

Thank you very much for your careful review and constructive suggestions. Those comments are all very valuable and helpful for improving our paper, as well as an important guiding significance to our research. We have studied the comments carefully and tried our best to revise the manuscript. Responses have been made point by point to the comments, and the modifications are highlighted **with red** in the revised Manuscript and revised Supplementary information. We hope that these modifications will make our manuscript qualified for publishing in your esteemed journal. The responses to the comments are detailed as follows.

Responses to Reviewer #1

Reviewer #1:

This is a review of the paper by Jin et al., on a cardiologist-level knowledge-fused Deep Learning model for arrhythmia diagnosis. The authors justify the need for such a model to help with automated electrocardiogram (ECG) readings, especially in the remote and underserved areas of China. The authors used data from January 2017 to December 2021, and studied to 272,753 Chinese patients. They achieved a 93.74% accuracy and reported that the diagnosis of 6 common arrhythmias using the model was better than performance of cardiologist. What is commendable is that the authors built a large clinical dataset consisting of single-lead ECG recordings of 10-45 seconds duration. The term the model as SJTU-ECGNet which was developed as a combination of a deep learning neural network (DNN), and medical diagnosis knowlede. To train the model 51,261 ECG recordings were used. The whole dataset was divided into 4 parts, including 40, 000 ECGs for training set, 5, 000 ECGs for verification set, 5, 000 ECGs for test set and 1, 261 ECGs for hidden set (cardiologist testing).

These are several concerns that exist:

Major:

Comment 1: The introduction section is quite lengthy and stretched out. There are details that are already mentioned in the methods section so these can be cut short.

Response 1: Thank you so much for your suggestion. We have re-examined the last paragraph of the introduction. The redundant details have been removed, and the clarity of the paragraph has been enhanced accordingly. **In the revised manuscript, we have modified the introduction Section, namely,**

"

Considering the application scenario of telemedicine, single-lead ECG acquisition is convenient and more suitable for out-of-hospital ECG diagnoses in remote and underdeveloped areas. This paper builds a large-scale, well-labelled clinical multi-label single-lead dataset consisting of various arrhythmias in China. The dataset contains 352,725 clinical recordings ranging from 10-45s in duration, and sourced from 272,753 different Chinese patients between January 2017 and December 2021. Meanwhile, over the course of more than a decade²⁰⁻²³, our team has diligently worked towards the wearable ECG acquisition device, accompanied by a mobile application with cloud-based AI diagnosis model to provide AI-aided solutions for out-of-hospital ECG diagnosis in remote and underdeveloped areas.

"

Comment 2: In data pre-processing, the authors describe undersampling to normalize all ECG recordings to a standard of 4096 sample points. It seems that this changes the width of some of the complexes. What are the challenges if model training is performed on raw samples with their length differences? Was model performance made better by the undersampling, or was that done to make it more intuitive to humans?

Response 2: Thank you for your insightful comments on our paper. Regarding the down-sampling method we used in the data pre-processing, it changes the width of some of the complexes. However, we also find that **it is a common and effective pre-processing method in the ECG classification field, which has been demonstrated in previous research^{13,14}**. We normalize all ECG recordings to a standard of 4096 sample points to achieve scale equivariance of deep features and to increase feature diversity, which does not change the diagnosis of the disease under study. For example, the critical features required for the atrial fibrillation and premature beats diagnosis, such as the RR intervals variability, remain unchanged before and after under-sampling. **Regarding the challenges of modelling training on raw samples with different lengths**, there are two main issues. Firstly, our proposed model structure requires the uniformization of sample lengths within each batch during training. Secondly, the large length differences in hospital data can introduce a potential data bias and redundant information that negatively impacts the extraction of information

by the RNN model, thus limiting the accuracy of the model. **Under-sampling can improve model performance, and other methods like padding or truncation may cause the loss of signal information and consequently limit model performance.** We appreciate your feedback and hope that our response addresses your concerns. **In the revised Supplementary information, we have added detailed discussion, namely**

"

According to the existing articles^{13,14}, under-sampling technique is a common and effective pre-processing method in the ECG classification field. Notably, we resample each recording $x_i^{t_i}$ to keep each input $\tilde{x}_i^{t_i}$ equal in length of 4096 sampling points, which does not change the diagnosis of the disease under study. For example, the critical features required for the atrial fibrillation and premature beats diagnosis, such as the RR intervals variability, remain unchanged before and after under-sampling. And then, under-sampling can improve model performance, and other methods like padding or truncation may cause the loss of signal information and consequently limit model performance.

13 S. Hong, W. Zhang, C. Sun, Y. Zhou, and H. Li. Practical Lessons on 12-Lead ECG Classification: Meta-Analysis of Methods From PhysioNet/Computing in Cardiology Challenge 2020. *Frontiers in Physiology*, 2021, 12: 811661.

14 Y. Bu, X. Cha, J. Zhu, Y. Su, D. Lai. Automatic detection model of hypertrophic cardiomyopathy based on deep convolutional neural network. *Journal of Biomedical Engineering*, 2022, 39(2): 285-292.

"

Comment 3: The authors used a neural network architecture consisting of 1D convolutional neural network (CNN), residual blocks, bidirectional long short-term memory (BiLSTM layer), and attention maps (ATT) for model explainability. Did the authors try a standard neural network model, before attempting more complex architecture?

Response 3: Thank you for taking the time to review our paper and providing your valuable feedback. We try a standard neural network model before implementing our proposed architecture.

We replicate ResNet-based model (Model e) from previous study on our dataset. The results in Table 1 and Table 2 indicate that the ResNet-based model achieved an accuracy of 86.68% on the test dataset and 73.83% on the Hidden dataset. In addition, we implement **the VGG model (Model f)** to classify ECG in the same configuration, achieving 89.60% and 76.84% accuracy on the test and hidden datasets, respectively. Hence, we propose SJTU-ECGNet model, which includes a 1D CNN, residual blocks, BiLSTM layer, and ATT, that achieving better accuracy in both the test and

hidden datasets compared to these standard neural network models. **In the revised manuscript, we have added detailed discussion, namely**

"

The results in Table 1 and Table 2 indicate that the ResNet-based model achieves an accuracy of 86.68% on the test dataset and 73.83% on the Hidden dataset. In addition, we implement the VGG model (Model f) to classify ECG in the same configuration, achieving 89.60% and 76.84% accuracy on the test and hidden datasets, respectively. Hence, the proposed SJTU-ECGNet model achieves better accuracy on both the test and hidden datasets compared to these standard neural network models.

"

And, we have added the details in the revised Supplementary information, namely

"

Supplementary Figure 9: Ablation experiments and comparison experiments models.

a, Ablation experiment without BiLSTM and ATT. b, Ablation experiment without ATT. c, Ablation experiment without BiLSTM. d, Comparison experiments: The proposed method without medical post-processing module. e, Comparison experiments: Ribeiro's method. f, Comparison experiments: Simonyan's method¹⁰.

"

Comment 4: What was the total computational time needed to train the ECGs for this study?

Response 4: Thank you for taking the time to review our paper and providing your valuable feedback. We train the proposed structure on the local workstation with NVIDIA GTX 1080 GPU. During the training process, we set batch size as 128. According to the statistical results, each epoch of training takes 67.79s, that is, each batch takes 0.22s, and each ECG recording takes 0.0017s to train. In future work, we will configure workstations with better performance, which will greatly reduce the time-consuming of training the model. Meanwhile, the model can be used directly after

training, which will ensure the real-time requirements of the model application. **In the revised Supplementary information, we have added detailed discussion, namely**

“

This paper trains the proposed SJTU-ECGNet structure on the local workstation with NVIDIA GTX 1080 GPU. Supplementary Table 4 shows the training cost of SJTU-ECGNet. During the training process, we set batch size as 128. According to the statistical results, each epoch of training takes 67.79s, that is, each batch takes 0.22s, and each ECG recording takes 0.0017s to train. In future work, we will configure workstations with better performance, which will greatly reduce the time-consuming of training the model. Meanwhile, the model can be used directly after training, which will ensure the real-time requirements of the model application.

	one epoch	one batch	one ECG recording
Time consumption	67.79 s	0.22 s	0.0017 s

Supplementary Table 4: Training time consumption of SJTU-ECGNet

”

Comment 5: There was a medical post-processing module after the initial neural network training. The authors mention the need for careful heart rate measurement, peak detection for RR intervals as a reason for this. These tasks have been shown to be possible with neural networks. Examples include peak prominence in Python or the various filters available in Matlab.

Response 5: Thank you for taking the time to review our paper and providing your valuable feedback regarding the medical post-processing module we used after the initial neural network training. We appreciate your suggestion of using neural networks for heart rate measurement and peak detection for RR intervals as an alternative. After conducting a thorough literature review, we utilize **a recent deep learning-based R wave localization (RPNet) algorithm** to implement medical post-processing. We compare this method with the Hamilton segmentation algorithm and find that this approach **is inferior to the Hamilton segmentation algorithm**. The comparison results on the hidden set are listed as followed:

Table:	Comparison results of Hamilton segmentation and RPNet in the hidden set						
	Normal	ST	SB	PAC	AF	PVC	Average
Hamilton	97.32	94.18	85.00	56.41	82.93	85.25	83.51
RPNet	85.89	92.63	85.00	56.41	82.93	85.25	80.44

As shown in the above Table, we conduct an analysis on these two models, the Hamilton segmentation algorithm has a better performance than deep learning-based R waves location algorithm. What's more, we visualize two cases to show the differences between two algorithms. As shown in Figure below, the yellow line represents that RPNNet ignores some R points during the point searching process, resulting in inaccurate RR feature calculation.

Generally, deep learning point searching algorithms do not exhibit border effects and have a wider range of point search. However, their algorithmic stability is inadequate. Therefore, the traditional Hamilton segmentation algorithm is more suitable for our research in this paper. **In the revised Supplementary information, we have added detailed discussion, namely**

"

After conducting a thorough literature review, we utilize a recent deep learning-based R wave localization (RPNNet) algorithm¹¹ to implement medical post-processing. We compare this method with the Hamilton segmentation algorithm and find that this approach is inferior to the Hamilton segmentation algorithm. The comparison results on the hidden set are listed as followed.

	Normal	ST	SB	PAC	AF	PVC	Average
Hamilton	97.32	94.18	85.00	56.41	82.93	85.25	83.51
RPNNet	85.89	92.63	85.00	56.41	82.93	85.25	80.44

Supplementary Table 1: Comparison results of Hamilton segmentation and RPNNet in the hidden set

As shown in the Supplementary Table 1, we conduct an analysis on these two models, the Hamilton segmentation algorithm has a better performance than deep learning-based R waves location algorithm. What's more, we visualize two cases to show the differences between two algorithms. As shown in Supplementary Figure 13, the yellow line represents that RpNet ignores some R points during the point searching process, resulting in inaccurate RR feature calculation.

Supplementary Figure 13: The visualization of two cases to show the differences between two algorithms

Generally, deep learning point searching algorithms do not exhibit border effects and have a wider range of point search. However, their algorithmic stability is inadequate. Therefore, the traditional Hamilton segmentation algorithm is more suitable for our research in this paper.

11 S. Vijayarangan, V. R., B. Murugesan, P. S.P., J. Joseph and M. Sivaprakasam. RpNet: A Deep Learning approach for robust R Peak detection in noisy ECG. 2020 42nd Annual International Conference of the IEEE Engineering in Medicine & Biology Society (EMBC), Montreal, QC, Canada, 2020, pp. 345-348, doi: 10.1109/EMBC44109.2020.9176084.

”

Comment 6: It seems cumbersome to include a post-processing module with clinician's reading thousands of ECGs. The question that arises is whether this step is even required? Why not trust the model to perform without the clinician post-processing? It appears, based on Table 1 that Model d that was without post-processing had the same AUROC as the SJTU-ECGNet in prediction of PAC,

AF, and PVC. So, is it justified to have clinicians go over thousands of ECGs to help differentiate sinus tachycardia from sinus bradycardia?

Response 6: Thanks for your valuable feedback, before answering this comment, we apologize for failing to provide sufficient clarification regarding the medical post-processing, which caused your misunderstanding.

Firstly, the post-processing module is an **automated process** judged by strict medical criteria without clinician involvement, which is presented in Supplementary Algorithm 1. In the whole diagnosis procedure, the clinicians are only involved in data rebelling process with auxiliary information including age and gender of each record to compare the clinical performance of our model in the hidden dataset.

Secondly, the medical post-process module **helps to correct the sinus rhythm results of DL-model**. In practice, DL-based model encounters performance degradation and confusion when classifies the arrhythmias that requires accurate parameters judgement. This is mainly because some common arrhythmia diseases, including ST and SB, depend on heart rate, RR interval and other indicators. These diagnostic indicators depend on the accurate position of the R-peaks. However, the DL-based model cannot accurately extract R-peaks, which will have a negative impact on the diagnostic effect of such arrhythmias. Therefore, we propose a rule-based model from medical prior knowledge. The specific algorithm is presented in Supplementary Algorithm 1.

Thirdly, according to Table 1, in ablation experiment, Model (d) that without post-process procedure had worse F1 score in the prediction of Normal, ST, SB than our SJTU-ECGNet, while same results in the prediction of PAC, AF and PVC. This outcome supports **our rule-based medical post-process module**, since **only sinus rhythm results were modified**, thus the predictions of PAC, AF and PVC would remain the same. Compared with the proposed method without medical post-processing module, the medical post-processing module can effectively correct the diagnostic errors of DNN in sinus arrhythmias, and effectively complement the DL-based model. **In the Discussion section of revised manuscript, we made some amendments and clarified some of our findings, namely**

"

Furthermore, the medical post-process module significantly improves the F1 score of sinus rhythm arrhythmias, including Normal, ST and SB. Table 1 reflects that in the test set, the F1 score of Normal increased from 94.78 to 97.74, for ST, it increases from 67.86 to 90.69, and for ST, it increases from 65.86 to 89.77. The experimental results highlight the effectiveness of the medical post-process module in enhancing the performance of detecting sinus rhythm arrhythmias.

"

Comment 7: The training data was exclusively made up of Chinese patients. How do the authors feel about clinical use and generalizability in centers across the world? Unless the model is validated in other datasets, this is an expected but significant issue that must be mentioned in the limitations section.

Response 7: Thank you for taking the time to review our paper and providing your valuable feedback. We appreciate your suggestion of model generalizability of the proposed SJTU-ECGNet. After conducting a thorough literature review, we use MIT-BIH dataset¹⁸, CODE dataset⁹, and PTB-XL dataset³¹ as external datasets for evaluating the model generalizability of the proposed SJTU-ECGNet. In addition, the ECG recordings in these three datasets are obtained from clinical settings that are completely different from the proposed dataset, with a high degree of independence. The experimental results are listed as follows.

	Normal	ST	SB	PAC	AF	PVC	F1-macro	Accuracy
MIT-BIH	94.47	-	-	65.77	93.69	91.83	86.44	91.07
CODE	99.61	97.33	94.30	-	96.83	-	97.02	99.21
PTB-XL	95.94	89.76	76.33	44.25	93.76	78.70	79.79	89.28

As shown in the above Table, SJTU-ECGNet can effectively extract the representative features from different datasets, and achieve 91.07%, 99.21%, 89.28% accuracy and 86.44%, 97.02%, 79.79% F1-macro in the three datasets, respectively. According to the existing research, the accuracy of ECG judgment by general clinicians is about 80%. The experimental results show that SJTU-ECGNet can achieve good model performance in different datasets. Therefore, the proposed network structure has good generalization ability. **In the Discussion section of Supplementary information, we have added the discussions, namely**

"

We use MIT-BIH dataset, CODE dataset, and PTB-XL dataset as external datasets for evaluating the model generalizability of the proposed SJTU-ECGNet structure. The ECG recordings in these three datasets are obtained from clinical settings that are completely different from the proposed dataset, with a high degree of independence. Supplementary Table 3 lists the experimental results.

	Normal	ST	SB	PAC	AF	PVC	F1-macro	Accuracy
MIT-BIH	94.47	-	-	65.77	93.69	91.83	86.44	91.07
CODE	99.61	97.33	94.30	-	96.83	-	97.02	99.21
PTB-XL	95.94	89.76	76.33	44.25	93.76	78.70	79.79	89.28

Supplementary Table 3: Comparison results of MIT-BIH, CODE and PTB-XL datasets

As shown in the Supplementary Table 3, SJTU-ECGNet can effectively extract the representative features from different datasets, and achieve 91.07%, 99.21%, 89.28% accuracy and 86.44%, 97.02%, 79.79% F1-macro in the three datasets, respectively. According to the existing research, the accuracy of ECG judgment by general clinicians is about 80%. The experimental results show that SJTU-ECGNet can achieve good model performance in different datasets. Therefore, the proposed network structure has good generalization ability.

9 Ribeiro AH, Ribeiro MH, Paixão GMM, et al. Automatic diagnosis of the 12-lead ECG using a deep neural network. *Nat Commun* 2020; 11: 1–9.

18 Goldberger AL, Amaral LAN, Glass L, et al. PhysioBank, PhysioToolkit, and PhysioNet: Components of a New Research Resource for Complex Physiologic Signals. *Circulation* 2000; 101: e215–20.

31 Wagner, P. et al. PTB-XL, a large publicly available electrocardiography dataset. *Sci Data* 7, 154 (2020).

"

Minor:

Comment 8: Did the authors try and analyze the ECG recordings of the 24.9% of patients that had their ECG misinterpreted by at least one of the three cardiologists, but with better performance by the model?

Response 8: Thank you for your thoughtful comments and for providing a new perspective on our study. Regarding the misinterpretation of ECG recordings, **we conduct an analysis of all 292 ECGs that are not interpreted correctly by all of the three cardiologists.** We then compare the diagnostic accuracy of the cardiologists and our model. The individual diagnostic accuracy of the three cardiologists is **37.33%, 18.49%, and 40.75%**, respectively, whereas the model's accuracy is **80.48%**. Based on these results, **we conclude that our model performed significantly better than the cardiologists in interpreting ECG recordings that are challenging for them.** Therefore, we believe that the proposed method has important implications for clinical practice, particularly in cases where the accuracy of ECG interpretation is critical. **In the revised Supplementary information, we have added detailed discussion, namely**

"

And then, regarding the misinterpretation of ECG recordings, we conduct an analysis of all 292 ECGs that are not interpreted correctly by all of the three cardiologists. We then compare the diagnostic accuracy of the cardiologists and our model. The individual diagnostic accuracy of the three cardiologists is 37.33%, 18.49%, and 40.75%, respectively, whereas the model's accuracy is

80.48%. Based on these results, we conclude that our model performed significantly better than the cardiologists in interpreting ECG recordings that are challenging for them. Therefore, we believe that the proposed method has important implications for clinical practice, particularly in cases where the accuracy of ECG interpretation is critical.

"

Comment 9: Multiple spelling and grammatical errors throughout the paper.

Response 9: Thank you for taking the time to review our paper and providing your valuable feedback. **According to your advice, in the revised manuscript, we have proofread our paper by carefully checking the manuscript and Supplementary information. Some corrections are presented below.**

- 1) Section Abstract: The word "**is an important means of**" has been modified as "**is crucial for**".
- 2) Section Abstract: The word "**broad**" has been revised as "**promising**".
- 3) Section Introduction: The phrase "common effective" has been revised as "**the most effective and straightforward**".
- 3) Section Introduction: Add the word "**are**".
- 4) Section Methods: The word "**result**" has been amended as "**results**".
- 5) Section Methods: The word "**length**" has been modified as "**lengths**".
- 6) Section Methods: Add the word "**an automated**".
- 7) Section Discussion: Add the word "**and lack sufficient disease category**".
- 8) Supplementary information Section Methods: The word "**firstly**" has been revised as "**first**".
- 9) Supplementary information Section Methods: Add the word "**to**".
- 10) Supplementary information Section Methods: The word "**requires**" has been amended as "**require**".
- 11) Supplementary information Section Methods: The word "**assist**" has been modified as "**assistance**".

Responses to Reviewer #2

Reviewer #2:

I congratulate the authors for putting in great efforts to make automated ECG diagnosis of the common arrhythmia, which may go a long way into helping patients in remote areas. I have a few minor comments.

Comment 1: The introduction can be rephrased to exclude details of the methods like – “This paper builds a well-labelled, large-scale clinical multi-label single-lead dataset with different arrhythmias of China for model training and evaluating, containing 352, 725 clinical recordings with 10-45s duration from 272, 753 different Chinese patients from January 2017 to December 2021. Then, we design the classification arrhythmia methods (SJTU-ECGNet) by combining deep neural network (DNN) and medical diagnosis knowledge.....”

Response 1: Thanks for your valuable feedback. **According to your suggestion, in the revised manuscript, we have excluded the details of the methods in the Introduction section and briefly introduce the proposed diagnostic system to pave the way for the detailed introduction of subsequent methods, namely**

”

This paper builds a large-scale, well-labelled clinical multi-label single-lead dataset consisting of various arrhythmias in China. The dataset contains 352,725 clinical recordings ranging from 10-45s in duration, and sourced from 272,753 different Chinese patients between January 2017 and December 2021. Meanwhile, over the course of more than a decade²⁰⁻²³, our team has diligently worked towards the wearable ECG acquisition device, accompanied by a mobile application with cloud-based AI diagnosis model to provide AI-aided solutions for out-of-hospital ECG diagnosis in remote and underdeveloped areas.

”

Comment 2: The description of the conceptual model of the diagnostic system is very neatly described but it would be great to include it in the methods section rather than introduction

Response 2: Thank you for your thoughtful comments. According to your valuable suggestion, we introduce the proposed diagnostic system and present the details of the proposed system in detail in the Methods section. **In the revised manuscript, we have revised the description, namely**

”

And then, this paper utilizes a deep neural network (DNN) along with medical diagnosis knowledge for building an accurate arrhythmia classification model (SJTU-ECGNet). Figure 1 illustrates the design of the SJTU-ECGNet arrhythmia classification model. Meanwhile, the diagnostic system can provide clinicians with direct evidence for review and visual inspection by simultaneously outputting the prediction results and the model's attention on each region of the input ECG, revealing the decision-making reasoning process of the model. As a result, it helps to reduce the variability of clinical decision-making. These neural networks are interdependent and complementary in reaching the final diagnosis.

Figure 1: Diagnostic system framework.

a, Demand and application for arrhythmia telemedicine. Considering the daily health monitoring and remote cardiovascular disease diagnosis in underdeveloped areas, a reliable and accurate automatic arrhythmia diagnosis system is urgently needed. b, The model structure of diagnostic system, including DL-based Module with several DNNs to output initial result and Medical Post-processing Module to obtain final result. The generated attention maps reveal the decision-making mechanism of clinicians.

"

Comment 3: The figures should be included in the chronological order in the manuscript e.g. Supplementary Figure 8 will come into the description after 1. Please change the numbers of figures wherever needed

Response 3: Thank you for your valuable comments. **According to your suggestion, in the revised Supplementary information, we have changed the numbers of figures.**

Comment 4: "The main challenge in implementing an automatic ECG diagnosis system in practice is the absence of a standardized and comprehensive ECG dataset. Nowadays, ECG recordings are mainly stored in image or PDF formats, which are unsuitable for training AI models. Furthermore, open-source ECG datasets collect few patients, cover fewer types of arrhythmias, and have no complete diagnostic notes. Therefore, with the assistance of Shanghai First People's Hospital, this paper constructs a large-scale Chinese ECG dataset (LSCP-ECGDS) for meeting the requirements of out-of-hospital ECG diagnosis." This doesn't need to be in the results section. The authors may consider including it in the methods under a separate heading may be

Response 4: Thank you for taking the time to review our paper and providing your valuable feedback. **According to your suggestion, in the revised manuscript, we have put the abovementioned paragraph into the Methods section, namely**

"

Methods

The main challenge in implementing an automatic ECG diagnosis system in practice is the absence of a standardized and comprehensive ECG dataset. Nowadays, ECG recordings are mainly stored in image or PDF formats, which are unsuitable for training AI models. Furthermore, open-source ECG datasets collect few patients, cover fewer types of arrhythmias, and have no complete diagnostic notes. Therefore, with the assistance of Shanghai First People's Hospital, this paper constructs a large-scale Chinese ECG dataset (LSCP-ECGDS) for meeting the requirements of out-of-hospital ECG diagnosis.

"

Comment 5: Kindly reword this statement with less redundancy – “For 1261 ECG recordings, 947(75.10%) recordings are all diagnosed correctly by three cardiologists and SJTU-ECG model, 97(7.70%) recordings are diagnosed correctly by two cardiologists and SJTU-ECG model, 33(2.62%) recordings are diagnosed correctly by one cardiologist and SJTU-ECG model and 105(8.33%) recordings are only diagnosed correctly by SJTU-ECG model.”

Response 5: Thanks for your valuable feedback. **According to your comments, in the revised manuscript, we have reworded this statement with less redundancy, namely**

"

Among the 1261 ECG recordings, 947 (75.10%) are accurately diagnosed by three cardiologists in accordance with the SJTU-ECGNet model, while 97 (7.70%) are misdiagnosed by the third cardiologist. Additionally, 33 (2.62%) recordings are accurately diagnosed by a single cardiologist and the SJTU-ECGNet model, and 105 (8.33%) recordings are solely diagnosed correctly by the SJTU-ECGNet model.

"

Comment 6: Discussion starts off with a good content but then the next 2 paragraphs again try to describe the results. It will be good to remove the duplication of wordings in the discussion section as compared to results and focus on the discussion on the salient features in the study per se so that the reader will understand the highlights of the study.

Response 6: Thank you for your thoughtful comments. **According to your valuable suggestion, in the revised manuscript, we remove some duplication of wordings in the discussion section and focus on the discussion on the highlights of the study.**

Comment 7: The paper needs proofreading to avoid incomplete statements like - “This paper describes the potential of automated AI-aided ECG diagnosis, which will bring greater convenience to clinical practice by fully developing.”

Response 7: Thank you for your valuable comments. **According to your comments, in the revised manuscript, we have proofread to avoid incomplete statements, namely**

”

This paper describes the potential of automated AI-aided ECG diagnosis, which will streamline screening processes and significantly contribute to reducing the workload of clinicians.

”

Comment 8: There are minor grammatical errors which may be helped by proofreading.

Response 8: Thank you for taking the time to review our paper and providing your valuable feedback. **According to your advice, in the revised manuscript, we have proofread our paper by carefully checking the manuscript and Supplementary information. Some corrections are presented below.**

- 1) Section Abstract: The word "**is an important means of**" has been modified as "**is crucial for**".
- 2) Section Abstract: The word "**broad**" has been revised as "**promising**".
- 3) Section Introduction: Add the word "**are**".
- 4) Section Methods: The word "**result**" has been amended as "**results**".
- 5) Section Methods: The word "**length**" has been modified as "**lengths**".
- 6) Section Methods: Add the word "**an automated**".
- 7) Section Discussion: Add the word "**and lack sufficient disease category**".
- 8) Supplementary information Section Methods: The word "**firstly**" has been revised as "**first**".
- 9) Supplementary information Section Methods: Add the word "**to**".
- 10) Supplementary information Section Methods: The word "**requires**" has been amended as "**require**".
- 11) Supplementary information Section Methods: The word "**assist**" has been modified as "**assistance**".

Responses to Reviewer #3

Reviewer #3:

The authors provide a new dataset and develop a model for classifying 6 types of arrhythmias.

I believe the general contribution is interesting, but I believe the manuscript is unclear in some key points.

There are also parts of the text that seem to be contradictory.

Major:

Comment 1: There seems to be some contradicting information: In the introduction, it is said that the dataset is a single lead, and this is the impression that I got from Figure 1. But in Supplementary Figure 8 it says that it is a 12-lead recording (and also in the method section).

Response 1: To ensure the feasibility of remote real-time diagnosis, the wearing convenience of sampling equipment needs to be considered. **Thus, this paper adopts the long-term monitoring of lead II ECG, which can not only ensure the preliminary diagnosis of common arrhythmia diseases, but also ensure the convenience of wearing the sampling equipment.** We have reorganized the figures in the Supplementary Information, resulting in Supplementary Figure 8 being relabeled as Supplementary Figure 1. **And in Supplementary Figure 1(a), we amend inappropriate label to improve the clarity, which highlights the lead II ECG.**

"

The revised Supplementary Table 1.a is shown below:

"

Comment 2: The data exclusion process is unclear. In the introduction and abstract, it mentions 272,753 Chinese patients. But in methods, it says that 51,261 ECGs are used in the development.

Maybe adding a diagram in the style of STARD diagram to report the flow of participants through the study could fix the problem.

Response 2: Thank you for valuable suggestions, it's indeed helpful to add a STARD diagram to figure out flow of participants through study. **And in the revised Supplementary information, we have added it in Supplementary Figure 1. Also, we have provided additional details in method, namely**

"

The STARD diagram of Supplementary Table 1.d is shown below:

A total of 51,261 lead II ECGs are collected for model training, validation, and hidden testing. Among these, 40,000 ECGs are allocated to the training dataset, 5,000 ECGs to the validation dataset, another 5,000 ECGs to the validation dataset, and 1261 ECGs to the hidden dataset. Initially, 352,725 clinical 12-lead ECGs are obtained from 272,753 patients. 29,315 ECGs are excluded due to poor signal quality. Additionally, 94,322 ECGs are excluded as they contained not only the six diseases of interest but also other types of arrhythmias. Furthermore, 177,827 ECGs are excluded to eliminate duplicates and for potential utilization in future research. A comprehensive description of the screening process is provided in Supplementary Figure 1(d).

"

Comment 3: The dataset is not properly described. Supplementary table 8 does some description. But the absolute numbers are absent. I believe this is a key piece of information that is missing. It is also unclear to be how many exams in each class there are. I believe a table would be significantly more transparent.

Response 3: Your suggestion is very helpful to us. Adding absolute numbers is crucial in multi-label classifications, as the total number of diseases may not match the total number of recordings. We have incorporated absolute numbers in Supplementary Figure 1(c) under the corresponding label to enhance clarity. In addition, Supplementary Figure 1(b) now includes the absolute numbers of sex and age to provide a more comprehensive presentation of the dataset information. **And in the revised Supplementary information, we have revised Supplementary Figure 1, namely**

''

Supplementary Figure 1: Organization for Large-scale Chinese people's ECG dataset (LSCP-ECGDS).

a, The 12 leads ECG measurement system, including 6 limb leads (3 pressurized limb leads and 3 bipolar limb leads) and 6 chest leads. Lead II is adopted in this paper considering the convenience of wearing the sampling equipment. b, Dataset distribution. The dataset comprises 51,261 clinical recordings from over 50,000 patients by carefully selecting from the whole LSCP-ECGDS, where 48.3% are male and 51.7% are female with ages covering the range from the young to the elder

(7.5% below 25 years, 39.2% from 25 to 50 years, 45.9% from 50 to 75 years and 7.4% over 75 years). The whole dataset is divided into 4 parts, including 40, 000 ECGs for training set, 5, 000 ECGs for verification set, 5, 000 ECGs for test set and 1, 261 ECGs for hidden set (cardiologist testing). Among them, there are 49, 579 recordings with one label, 1, 642 with two labels, and 40 with three labels. c, The six common arrhythmias consist of 81.8% normal cases and 18.2% abnormal cases. Within the abnormal category, the following subcategories are present: ST (41%), SB (35%), PAC (10%), PVC (8%), and AF (6%). d, STARD (Standards for Reporting of Diagnostic Accuracy Studies) flow diagram of LSCP-ECGDS. 29,315 ECGs are excluded due to poor signal quality. 94,322 ECGs are excluded as they contain not only the six diseases of interest but also other types of arrhythmias. 177,827 ECGs are excluded to eliminate duplicates and for potential utilization in future research. The whole dataset is divided into 4 parts, including 40, 000 ECGs for training set, 5, 000 ECGs for validation set, 5, 000 ECGs for test set and 1, 261 ECGs for hidden set (cardiologist testing). Among them, there are 49, 579 recordings with one label, 1, 642 with two labels, and 40 with three labels.

"

Comment 4: It is unclear from the text why the author divides into two different test sets. What is exactly the role of the Hidden test set? In Supplementary table 8, it says that it is for Doctor testing. But it is not clear to me why the division.

Response 4: We appreciate your valuable feedback and would like to explain the purpose of the two different test sets in our research. In our study, **the test set is a commonly used dataset in deep learning used to evaluate the model's performance, such as accuracy.** However, **the hidden set is comprised of 1261 ECG recordings that are independently labelled by three clinical cardiologists who reviewed every record one by one, like taking examination.** Hence, **the main goal of using the hidden set is to calculate the actual diagnostic accuracy of clinical cardiologists.** Even clinical cardiologists can make mistakes when interpreting ECGs. **By using this hidden set, we aim to compare the diagnostic performance differences between our SJTU-ECGNet model and three clinical cardiologists, which result in evaluating the effectiveness of our model in clinical screening. And in the revised manuscript, we have added the description, namely**

"

Further, the test set is used to evaluate the model's performance, such as accuracy. Additionally, the hidden set is comprised of 1261 ECG recordings that are independently labelled by three clinical cardiologists. The hidden set is used to compare the diagnostic performance differences

between our SJTU-ECGNet model and three clinical cardiologists, which result in evaluating the effectiveness of our model in clinical screening.

"

Comment 5: Why not compute the F1 score and accuracy of the doctors?

Response 5: Thank you for taking the time to review our paper and providing your comments. We appreciate your feedback and have carefully considered your suggestion regarding the computation of F1 score and accuracy of the cardiologists. In our paper, we focus on comparing and analysing the diagnostic accuracy and F1-macro of our model with that of the cardiologists. We find that our model achieves an accuracy of 85.49%, which is higher than the accuracy of three of the cardiologists (81.13%, 86.28%, and 93.74%). **Therefore, our model outperforms the cardiologists and exhibits a better diagnostic accuracy than them.** Regarding the F1-macro, we calculate it for both the cardiologists and the model. **The experimental results indicate that the F1-macro of the cardiologists ranged from 79.40% to 92.88%, while our model achieves an F1-macro of 83.51%, which demonstrates a similar level of performance to that of the cardiologists.** In conclusion, our study indicates that the model outperforms most of the cardiologists in terms of accuracy and demonstrates comparable performance in terms of F1-macro. **In the revised Supplementary information, we have revised the Results section and Discussion section, namely**

"

Additionally, we focus on comparing and analysing the diagnostic accuracy and F1-macro of our model with that of the cardiologists. As shown in the Supplementary Figure 10, We find that our model achieves an accuracy of 93.74%, which is higher than the accuracy of three of the cardiologists (81.13%, 86.28%, and 85.49%). Therefore, our model outperforms the cardiologists and exhibits a better diagnostic accuracy than them. Regarding the F1-macro, we calculate it for both the cardiologists and the model. The experimental results indicate that the F1-macro of the cardiologists ranged from 79.40% to 92.88%, while our model achieves an F1-macro of 83.51%, which demonstrates a similar level of performance to that of the cardiologists.

In conclusion, our study indicates that the model outperforms most of the cardiologists in terms of accuracy and demonstrates comparable performance in terms of F1-macro.

Supplementary Figure 10: Comparison of three cardiologists and SJTU-ECGNet in diagnostic accuracy and F1-macro in the re-label hidden set.

”

Comment 6: I think it would be interesting if the authors also provided a table with other traditional metrics (Specificity, sensitivity, PPV, NPV)

Response 6: Thank you very much for your detailed and insightful comments on our paper. We appreciate your suggestion to include a table with other traditional metrics like specificity, sensitivity, PPV, and NPV. We have performed additional calculations to obtain these metrics and have included them in our revised manuscript.

Specifically, we calculate these metrics for both our model and the three cardiologists on the hidden set. As shown in the Table below, **our model achieves better results in terms of specificity and NPV, while achieving comparable results with the cardiologists in terms of sensitivity and PPV.** The results are shown as follows.

Table:	Four metrics comparison results of SJTU-ECGNet and cardiologists on the hidden set			
	Sensitivity	Specificity	PPV	NPV
SJTU-ECGNet	81.86	98.50	86.31	98.52
Cardiologist1	85.21	93.50	91.67	98.20
Cardiologist2	74.00	90.61	89.48	97.32
Cardiologist3	92.97	93.48	93.01	98.48

In the revised Supplementary information, we have revised the Discussion section, namely

”

Specifically, we calculate these metrics for both our model and the three cardiologists on the hidden set. As shown in the Supplementary Table 2, SJTU-ECGNet achieves better results in terms of specificity and NPV, while achieving comparable results with the cardiologists in terms of sensitivity and PPV.

	Sensitivity	Specificity	PPV	NPV
SJTU-ECGNet	81.86	98.50	86.31	98.52
Cardiologist1	85.21	93.50	91.67	98.20
Cardiologist2	74.00	90.61	89.48	97.32
Cardiologist3	92.97	93.48	93.01	98.48

Supplementary Table 2: Four metrics comparison results of SJTU-ECGNet and cardiologists on the hidden set

..

Comment 7: I found Figure 2 quite unclear. It is hard to read the results from it.

Response 7: We appreciate your valuable suggestion on our manuscript. **Figure 2 compares diagnostic performance between SJTU-ECGNet and 3 different cardiologists.** After carefully analyzing your comment, we agree that **Figure 2 (a) should be improved as it presents the diagnostic bias matrix in a slightly obscure way.** In the Figure 2 (a), **diagnostic inconsistency between cardiologists and the ECG model, as well as their inconsistency with the ground truth label,** are expressed using an upper triangular matrix. Each element in the matrix corresponds to a pair of objects, with the numerical value indicating their degree of inconsistency. To clarify the presentation of this figure, we redefine it by using **Hamming Loss** to express the diagnostic bias between any two objects. The values in each position of the matrix are expressed as follows:

$$M_{ij} = \frac{1}{mq} \sum_{k=1}^m \sum_{n=1}^q I(y_{i,n}^{(k)} \neq y_{j,n}^{(k)})$$

Here, i and j represent the position in the matrix, m is the samples number and q is the classes number. For example, $M(1,2)$ indicates the diagnostic bias between cardiologist1 and cardiologist2 while interpreting the ECGs. The lower the value, the more consistent the diagnosis between the two doctors. Through analysis, it can be determined that there is subtle inconsistency in diagnosis among cardiologists. Specifically, the inconsistency between cardiologist 1 and 2, cardiologist2 and cardiologist3, and cardiologist1 and cardiologist3 are 0.03, 0.016, and 0.027, respectively. However, the difference from the true labels is significantly larger than the internal inconsistencies, with differences of 0.029, 0.043, and 0.026, respectively. Conversely, the ECG

model demonstrates a higher level of consistency with the true labels, with a score of 0.015, which reflects better consistency than cardiologists.

Figure 2 (b) illustrates the accuracy of each recording in the hidden set diagnosed by our model and three cardiologists respectively. We agree that the absence of numerical representations and labels in the figure made it less intuitive and difficult to understand. **Therefore, we have added numeric values to the figure to improve its readability and clarity. The colorful circle dots represent distinct diagnostic outcomes generated by our SJTU-ECGNet model and three cardiologists. For instance, the light blue dots indicate that 947 recordings are accurately diagnosed by all three cardiologists and the SJTU-ECG model, and the black dots represent 97 recordings are correctly diagnosed by the model and two cardiologists but are misdiagnosed by the third cardiologist, and so on.** The '✓' mark denotes the correctly classified recordings, meaning that the predicted label set is identical to the ground-truth label set. The '×' mark indicates that the predicted label set is either partially or entirely incorrect compared to the ground-truth label set.

We hope our efforts to improve Figure 2 have addressed the reviewer's concerns and contributed to the clarity of our manuscript. **In the revised manuscript, we have added detailed discussion, namely**

"

To further compare the diagnostic effectiveness of SJTU-ECGNet with that of cardiologists, diagnostic inconsistency matrix and diagnostic correctness charts are respectively proposed in Figure 2 (a) and Figure 2 (b). There exist differences in the diagnostic inconsistency between different cardiologists, the inconsistency of cardiologist1 and cardiologist3 is 0.016, the inconsistency of cardiologist1 and cardiologist2 is 0.03, the inconsistency of cardiologist2 and cardiologist3 is 0.027. Compared to the ground-truth label, the inconsistency of three cardiologists is respectively 0.029, 0.043, and 0.026, while the inconsistency of SJTU-ECGNet is 0.015, much smaller than that of cardiologists, which reflects better consistency than cardiologists. Among the 1261 ECG recordings, 947 (75.10%) are accurately diagnosed by three cardiologists in accordance with the SJTU-ECGNet model, while 97 (7.70%) are misdiagnosed by the third cardiologist. Additionally, 33 (2.62%) recordings are accurately diagnosed by a single cardiologist and the SJTU-ECGNet model, and 105 (8.33%) recordings are solely diagnosed correctly by the SJTU-ECGNet model.

Figure 2: Comparison of SJTU-ECGNet and cardiologists in diagnosis effect

a. Diagnostic inconsistency matrix between cardiologists and SJTU-ECGNet. (Using Hamming loss to calculate inconsistency between a pair of objects) b. Diagnostic correctness charts in each ECG recording (total 1261 ECG recordings). For instance, the light blue dots indicate that 947 recordings are accurately diagnosed by all three cardiologists and the SJTU-ECG model, and the black dots represent 97 recordings are correctly diagnosed by the model and two cardiologists but are misdiagnosed by the third cardiologist, and so on. The ‘✓’ mark denotes the correctly classified recordings, meaning that the predicted label set is identical to the ground-truth label set. The ‘×’ mark indicates that the predicted label set is either partially or entirely incorrect compared to the ground-truth label set.

”

Comment 8: It is not clear how the threshold was selected. It is also unclear to me if a softmax or a sigmoid is being used (Sup. Figure 6 and 9 seem to contradict each other).

Response 8: Thank you very much for your detailed and insightful comments on our paper. Firstly, we apologize for the unclear statement on the selection of the threshold in our manuscript. To clarify, as we are dealing with a multi-label classification problem, we could not use the Softmax classifier that is typically used in multi-class problems, we use the sigmoid function as the classifier for our multi-label classification problem. As stated in the literature reference¹², the commonly used fixed threshold for multi-label problems is 0.5. Therefore, we choose the threshold of 0.5 for our classification.

Regarding the confusion between Sup. Figure 6 and 9, thank you for bringing it to my attention. Upon careful inspection, we find that there is an error in the plotting process for Figure 9. We have rectified the mistake and can confirm that it now aligns with the rest of the paper.

In the revised Supplementary information, we have added detailed discussion, namely

"

As stated in the literature reference¹², the commonly used fixed threshold for multi-label problems is 0.5. Therefore, we chose the threshold of 0.5 for our multi-label classifier.

12 Moyano J M, Gibaja E L, Cios K J, et al. Review of ensembles of multi-label classifiers: models, experimental study and prospects. Information Fusion, 2018, 44: 33-45.

Supplementary Figure 9: Ablation experiments and comparison experiments models.

a, Ablation experiment without BiLSTM and ATT. b, Ablation experiment without ATT. c, Ablation experiment without BiLSTM. d, Comparison experiments: The proposed method without medical post-processing module. e, Comparison experiments: Ribeiro's method. f, Comparison experiments: Simonyan's method¹⁰.

"

Minor:

Comment 9: In the Methods description, too much space is devoted to describing well-known performance metrics. And the description of the implementation details is very short.

Response 9: We appreciate your valuable suggestion. As per your recommendation, we have deleted unnecessary well-known formula of performance metrics. Additionally, we have included more detailed parameter setups in the implementation details. Furthermore, we have provided a comprehensive description of the neural network architecture in the Method section. **In the revised manuscript, we have added the details, namely**

"

Neural network architecture

Although the change in ECG is minimal, even subtle changes can have a significant impact on the result. This paper designs CNNs, Residual blocks and BiLSTM layer for automatically extracting features from ECGs, as shown in Supplementary Figure 4. The Residual block is responsible for extracting the short-term dependence features, while the BiLSTM layer extracts the long-term dependence features of ECG. Then, ATT layer is utilized to enhance beneficial features and indicates the model's focus on the input features. Finally, multi-label classifier processes the deep features to generate the prediction results. Following the DL-based prediction results, the medical post-processing module corrects some arrhythmia disease prediction results, including ST and SB. Combining DL-based model and medical post-processing module, we use the ECG characteristics of different arrhythmias to improve the final diagnostic accuracy. Besides, the details of arrhythmias diagnosis model are listed in Supplementary Figure 5-8.

Performance metric

To comprehensively evaluate the formidability and correctness of the model, this paper uses accuracy (Acc), macro F1 score (F1-macro), and the area under the curve of the receiver operating characteristic (AUC-ROC) as performance metrics²⁰. The F1-macro evaluates the overall performance of the model and is robust to data imbalance effects. The detailed formulas can be found in Supplementary Information.

The confidence interval (CI) was derived from bootstrap distribution, that is, to estimate parameters using a repeated sampling method. The 95% CI can be expressed as Equation (1).

$$(\hat{\theta} - t_{b,0.025}^* \times \hat{\sigma}_{\hat{\theta}}, \hat{\theta} + t_{b,0.025}^* \times \hat{\sigma}_{\hat{\theta}}) \quad (1)$$

Implementation details

This paper utilizes NVIDIA GTX 1080 GPU to train the proposed AI-aided model and applies Adam optimizer²⁸ to update network weights during backpropagation process. The learning rate is set to 0.001. Then, during the training process, Focal Loss is selected as loss function with

parameter $\gamma=2$. The model weights are initialized with kaiming uniform. The batch size is set as 128. The whole training process runs for 50 epochs, and the final model is selected based on the best validation results obtained during the optimization process.

"

Comment 10: In the discussion, the authors mention only a small size open-source ECG dataset. But there are some large-scale open-source datasets. I mention

PTB-XL(<https://doi.org/10.1038/s41597-020-0495-6>),

CODE (<https://doi.org/10.17044/scilifelab.15169716> - available for research upon request) and

CODE 15% (<https://doi.org/10.5281/zenodo.4916206>)

Response 10: Thanks for your suggestion, we have re-considered these two datasets. Although there are some large-scale open-source datasets like CODE and PTB-XL, **they are not suitable for this study due to constraints such as the absence of samples collected from the Chinese population and the restricted disease category in the CODE dataset that excludes PAC and PVC. In the Method section of revised manuscript, we have added the description, namely**

"

Other large-scale datasets like CODE⁹ and PTB-XL³¹ are not suitable for this study due to constraints such as the absence of samples collected from the Chinese population and the restricted disease category in the CODE dataset that excludes PAC and PVC.

³¹ Wagner, P. et al. PTB-XL, a large publicly available electrocardiography dataset. *Sci Data* 7, 154 (2020).

"

And then, in the Discussion section of revised manuscript, we have added the description, namely

"

On the other hand, some previous works have obtained data sets of a certain size from heart monitors and Holter tests, such as CODE⁹, CinC Challenge 2017³⁰ and PTB-XL³¹, but these datasets either do not contain multiple arrhythmias coupling conditions¹⁰ or lack disease categories of interest. Consequently, they do not fulfil the requirements for an end-to-end diagnostic model for clinical needs.

"

We hope these modifications will make our paper qualified for publication in your esteemed journal. I really appreciate your time and effort that goes into the processing of our manuscript.

Yours sincerely,

Professor Dr Chengliang Liu

State Key Laboratory of Mechanical System and Vibration, School of Mechanical Engineering,
Shanghai Jiao Tong University, Shanghai 200240, China

E-mail: chlliu@sjtu.edu.cn

Reviewers' comments:

Reviewer #2 (Remarks to the Author):

I'm satisfied with the current version of the manuscript and I have no more comments.

Reviewer #3 (Remarks to the Author):

I thank the reviewers for the response and for the clarification. I think in general the authors did a good job addressing my comments.

- On response 2, what does it mean duplicated patients? Does it mean the ECG belongs to the same patient? Most of the exams are excluded in this step, so I think it is important it is clear.

Also, Please make a clear distinction between the number of ECGs and the number of patients in each step in the figure and text.

- In response 5, claims of the type "our model outperforms the cardiologists" should be avoided unless statistical significance is established.

- In response 7, thanks for the clarification. I think I understand it now. But still, I find it very hard to get insight from it, would probably delegate it to the supplementary material

- In response 10, I would change the phrase: "On the other hand, some previous works have obtained data sets of a certain size from heart monitors and Holter test, such as CODE, ...". Because all these studies are from standard 12 leads ECGs of 10 seconds or so. Not heart monitors or Holter tests.

{ED: This reviewer also looked at your response to Reviewer 1, who was unavailable, and commented that the issues raised by this reviewer were adequately addressed.}

Summary of Responses and Amendments

Manuscript ID: **COMMSMED-23-0107A**

Manuscript Title: **Cardiologist-level interpretable knowledge-fused deep neural network for automatic arrhythmia diagnosis**

Journal: **Communications Medicine**

Dear Reviewers,

Thank you very much for your careful review and constructive suggestions. Those comments are all very valuable and helpful for improving our paper, as well as an important guiding significance to our research. We have studied the comments carefully and tried our best to revise the manuscript. Responses have been made point by point to the comments, and the modifications are highlighted **with red** in the revised Manuscript and revised Supplementary information. We hope that these modifications will make our manuscript qualified for publishing in your esteemed journal. The responses to the comments are detailed as follows.

Responses to Reviewer #2

Reviewer #2:

I'm satisfied with the current version of the manuscript and I have no more comments.

Response 1: Thank you for your review and suggestions on this paper.

Responses to Reviewer #3

Reviewer #3:

I thank the reviewers for the response and for the clarification. I think in general the authors did a good job addressing my comments.

Comment 1: On response 2, what does it mean duplicated patients? Does it mean the ECG belongs to the same patient? Most of the exams are excluded in this step, so I think it is important it is clear. Also, Please make a clear distinction between the number of ECGs and the number of patients in each step in the figure and text.

Response 1: Thank you for your valuable comments. We apologize for the lack of clarity in our description of the Hospital dataset.

Firstly, “duplicated patients” is due to the Hospital dataset comes from five years of clinical data. A patient may have ECG measured regularly during this period, or the doctor may require multiple ECG measurements during the diagnosis, resulting in a patient having several ECGs in the dataset. In this paper, in order to ensure a strict inter-patient experimental setting, we ensure that every ECG belongs to different patient.

Secondly, in the STARD figure (Supplementary Figure 1d), 177,827 records were excluded due to two reasons: duplicate patients and future research' use. This is because, during the initial construction of the Hospital dataset, part of ECGs was in PDF format, owing to the multi-center storage format of the Hospital data. Recently, we obtained this portion of ECGs in the same format as the data used in this study, and we use it as an external dataset to validate the performance of our model. Finally, we have modified the STARD figure to provide detailed information on the number of ECGs and patients involved in each step of the division.

And, we have added the description in the revised Supplementary information, namely

“

Supplementary Figure 1d: STARD (Standards for Reporting of Diagnostic Accuracy Studies) flow diagram of LSCP-ECGDS.

Comparison results of different datasets

We test the generalization of the proposed model on the external Hospital dataset and use MIT-BIH dataset, CODE dataset and PTB-XL dataset as external datasets for evaluating the model generalizability of the proposed SJTU-ECGNet structure. The ECG recordings in last three datasets

are obtained from clinical settings that are completely different from the proposed dataset, with a high degree of independence. Supplementary Table 3 lists the experimental results.

	Normal	ST	SB	PAC	AF	PVC	F1-macro	Accuracy
External Hospital	96.80	86.54	85.65	69.21	79.04	76.00	82.21	93.29
MIT-BIH	94.47	-	-	65.77	93.69	91.83	86.44	91.07
CODE	99.61	97.33	94.30	-	96.83	-	97.02	99.21
PTB-XL	95.94	89.76	76.33	44.25	93.76	78.70	79.79	89.28

Supplementary Table 3: Comparison results of external Hospital datasets, MIT-BIH, CODE and PTB-XL

As shown in the Supplementary Table 3, the proposed SJTU-ECGNet has good generalization, and also achieves 93.29% accuracy and 82.21% F1-macro on the external Hospital dataset, which is equivalent to the performance on the test set and hidden set, without overfitting. What's more, SJTU-ECGNet can also effectively extract the representative features from different datasets, and achieve 91.07%, 99.21%, 89.28% accuracy and 86.44%, 97.02%, 79.79% F1-macro in the last three datasets, respectively. According to the existing research, the accuracy of ECG judgment by general clinicians is about 80%. The experimental results show that SJTU-ECGNet can achieve good model performance in different datasets. Therefore, the proposed network structure has good generalization ability.

"

Comment 2: In response 5, claims of the type "our model outperforms the cardiologists" should be avoided unless statistical significance is established.

Response 2: Thank you very much for your valuable comments. We completely agree with your question about the rigorous wording of the article. We have modified this statement to indicate that the model is better than cardiologists in diagnosing accuracy in the hidden set. We have also made corrections to other similar statements in the article. **In the revised manuscript, we have modified the statement in Results, namely**

"

Despite similar academic backgrounds or institutional affiliations, the diagnosis results on the hidden set vary considerably among cardiologists.

"

And, we have modified the statement in the revised Supplementary information, namely

"

Therefore, our model exhibits a cardiologist-level diagnostic accuracy on the hidden set.

"

Comment 3: In response 7, thanks for the clarification. I think I understand it now. But still, I find it very hard to get insight from it, would probably delegate it to the supplementary material

Response 3: Thank you very much for your comments. We appreciate your observation that the two figures in Figure 2 may not provide quick insights. In response to this concern, we have added annotations in the figures to highlight the areas that require specific attention. Additionally, we have included explanations in the legend to clarify the insights depicted in the figures. Furthermore, in the supplemental material, we have provided detailed explanations of the data calculation principles and formulas used in the figures. **In the revised manuscript, we have added information in Figure 2, namely**

"

We use hamming loss to calculate the diagnostic inconsistency between the model and cardiologists (The detailed calculations can be found in Supplementary Information Results).

Figure 2: Comparison of SJTU-ECGNet and cardiologists in diagnosis effect

a. Diagnostic inconsistency matrix between cardiologists and SJTU-ECGNet. (As the value decreases, the diagnostic consistency between the two increases. The smallest value is marked with a red box.) b. Diagnostic correctness charts in each ECG recording (total 1261 ECG recordings). The dots in polygonal region indicate that the performance of the model is comparable to, or even higher than, that of the cardiologists. The dots outside the region represent correct diagnoses made by cardiologists, but missed diagnosis or misdiagnosis by the model. For instance, the light blue dots indicate that 947 recordings are accurately diagnosed by all three cardiologists and the SJTU-

ECG model, and the black dots represent 97 recordings are correctly diagnosed by the model and two cardiologists but are misdiagnosed by the third cardiologist, and so on. The '✓' mark denotes the correctly classified recordings, meaning that the predicted label set is identical to the ground-truth label set. The '×' mark indicates that the predicted label set is either partially or entirely incorrect compared to the ground-truth label set.

"

And, we have added the definition of Diagnostic consistency in the revised Supplementary information, namely

"

Diagnostic consistency

We choose **Hamming Loss** to express the diagnostic bias between any two objects. The values in each position of the matrix in Figure 2(a) are expressed as follows:

$$M_{ij} = \frac{1}{mq} \sum_{k=1}^m \sum_{n=1}^q I(y_{i,n}^{(k)} \neq y_{j,n}^{(k)}) \quad (20)$$

Here, i and j represent the position in the matrix, m is the samples number and q is the classes number. For example, $M(1,2)$ indicates the diagnostic bias between cardiologist1 and cardiologist2 while interpreting the ECGs. The lower the value, the more consistent the diagnosis between the two doctors. Through analysis, it can be determined that there is subtle inconsistency in diagnosis among cardiologists.

"

Comment 4: In response 10, I would change the phrase: "On the other hand, some previous works have obtained data sets of a certain size from heart monitors and Holter test, such as CODE, ...". Because all these studies are from standard 12 leads ECGs of 10 seconds or so. Not heart monitors or Holter tests.

Response 4: Thank you very much for your suggestion. We have reconfirmed these large-scale open-source ECG datasets, among which the 12-lead ECG data of the CODE dataset is 7-10s, the PTB-XL dataset is all 10s, and the single-lead ECG data of CinC Challenge 2017 is 30-60s. **Therefore, according to your suggestion, in the revised manuscript, we have modified the statement in Results, namely**

"

On the other hand, some previous works have obtained datasets of a certain size from 12 leads ECGs or hand-held device, such as CODE⁹, CinC Challenge 2017³⁰ and PTB-XL³¹, but these

datasets either do not contain multiple arrhythmias coupling conditions¹⁰ or lack disease categories of interest.

"

We hope these modifications will make our paper qualified for publication in your esteemed journal. I really appreciate your time and effort that goes into the processing of our manuscript.

Yours sincerely,

Professor Dr Chengliang Liu

State Key Laboratory of Mechanical System and Vibration, School of Mechanical Engineering,
Shanghai Jiao Tong University, Shanghai 200240, China

E-mail: chlliu@sjtu.edu.cn

REVIEWERS' COMMENTS:

Reviewer #3 (Remarks to the Author):

The authors have addressed my concerns

Summary of Responses and Amendments

Manuscript ID: **COMMSMED-23-0107B**

Manuscript Title: **Cardiologist-level interpretable knowledge-fused deep neural network for automatic arrhythmia diagnosis**

Journal: **Communications Medicine**

Dear Reviewers,

Thank you very much for your careful review and constructive suggestions. Those comments are all very valuable and helpful for improving our paper, as well as an important guiding significance to our research. We hope that these modifications will make our manuscript qualified for publishing in your esteemed journal. The responses to the comments are detailed as follows.

Responses to Reviewer #3

Reviewer #3:

The authors have addressed my concerns.

Response 1: Thank you for your review and suggestions on this paper.

We hope these modifications will make our paper qualified for publication in your esteemed journal. I really appreciate your time and effort that goes into the processing of our manuscript.

Yours sincerely,

Professor Dr Chengliang Liu

State Key Laboratory of Mechanical System and Vibration, School of Mechanical Engineering,
Shanghai Jiao Tong University, Shanghai 200240, China

E-mail: chlliu@sjtu.edu.cn